Public domain. CC0 1.0.




# Topographic controls on landslide mobility: Modeling hurricane-induced landslide runout and debris-flow inundation in Puerto Rico

Dianne L. Brien[1], Mark E. Reid[1], Collin Cronkite-Ratcliff[2], Jonathan P. Perkins[2]

[1]U.S. Geological Survey, Volcano Science Center, Moffett Field, CA, 94035, USA

[2]U.S. Geological Survey, Geology, Minerals, Energy, and Geophysics Science Center, Moffett Field, CA, 94035, USA

*Correspondence to*: Dianne L. Brien (dbrien@usgs.gov)

**Abstract.** In 2017, Hurricane Maria triggered more than 70,000 landslides in Puerto Rico. After initiation, these predominantly shallow landslides mobilized to varying degrees – some landslides only traveled partway downslope, whereas others reached drainages and mobilized into long-traveled debris flows that could severely impact roads and infrastructure. Thus, forecasting

potential landslide runout and inundation zones is critical for estimating landslide and debris flow hazards. Here we conduct an in-depth topographic analysis of landslide-affected areas from nine study areas and apply a linked modeling technique to estimate locations susceptible to varying degrees of landslide runout in Lares, Utuado and Naranjito municipalities.

We find that longer runout length is observed on high-relief escarpments, although highly mobile long-runout debris flows also occurred in lower-relief dissected uplands. These topographic differences indicate that landslides initiating under

similar conditions and possessing equal potential to mobilize as debris flows may not travel the same distances or affect the same areal extent. Our modeling approach allows the local topography to automatically control the implementation of two runout methods: 1) *H/L* runout zones are assigned directly downslope of landslide source zones, and 2) debris-flow inundation zones are estimated in the presence of a channel network. Debris-flow volumes are calculated as a function of area-integrated growth factors, estimated as a function of the upstream areas susceptible to shallow landslides. Applying our empirical modeling scheme

over an area of 560 km$^2$, our results highlight the efficacy of our methods for assessment of the potential for landslide runout and debris-flow inundation over diverse terrains with varied susceptibility.

## 1 Introduction

Globally, 55,997 fatalities due to non-seismically triggered landslides were recorded over the twelve-year period between January 2004 and December 2016 (Froude and Petley, 2018). When conditions for landslide mobilization exist, including at least

partial liquefaction by high pore pressures, landslides may mobilize to form debris flows, fast-moving slurries of saturated, poorly sorted sediment (e.g., Iverson, 1997; Hungr et al., 2002). Fast-moving, far-traveled landslides, such as debris flows, are one of the most destructive types of landslides. Due to their rapid velocity and occurrence without warning, debris flows can be lethal (e.g., Highland and Bobrowsky, 2008; McDougall, 2017). In regions where humans and infrastructure are present, landslide susceptibility forecasting tools to identify potential runout zones for high-mobility landslides are of foremost

importance.

Landslide susceptibility models typically focus on a single type of landslide or process of movement, either landslide initiation (e.g., Montgomery and Dietrich, 1994; Larsen and Parks, 1998; Pack et al., 1999; Baum et al., 2008; Lepore et al., 2012; Mergili et al., 2014; Reid et al., 2015; Merghadi et al., 2020; Hughes and Schulz, 2020) or runout (e.g., Guzzetti at al., 2006; McDougall, 2017). Runout models may be empirically (e.g., Iverson et al., 1998; Horton et al., 2013; Berti and Simoni,

2014) or physics-based (e.g., McDougall and Hungr, 2004; Christen et al., 2010; George and Iverson, 2014; Iverson and George,

Public domain. CC0 1.0.

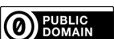



2014; FLO-2D Software Inc., 2007; Gorr et al., 2022) and are often focused on back-analysis or site-specific investigations (e.g., McDougall, 2017), typically requiring detailed information about location of landslide initiation and volume or a flow hydrograph (e.g., Barnhart et al., 2021). Empirical runout methods, based on power-law volume/area relations, such as Laharz (Iverson et al., 1998; Schilling, 2014) or DFLOWZ (Berti and Simoni, 2014), provide methods for automated delineation of

inundation areas of lahars (e.g., Major et al., 2004; Muñoz-Salinas et al., 2009) or debris flows (Crosta et al., 2002; Griswold and Iverson, 2008; Magirl et al., 2010). Several previous investigations have combined landslide models to estimate both landslide source (initiation) and runout zones. These investigations incorporated empirical models (Guinau et al., 2007; Mergili et al., 2019), physics-based models (Hsu and Liu, 2019), or a combination of empirical and physics-based methods (Ellen et al., 1993; Benda et al., 2007; Bregoli et al., 2010; Park et al., 2016; Fan et al., 2017; Pollock et al., 2019). However, existing methods for

analyzing runout do not directly account for location within the topography and the transition from non-channelized to channelized runout.

We build a conceptual framework to define zones of mobility within the landscape that provides the basis of our topographic analysis of landslide-affected areas (source and runout) and modeling approach in Puerto Rico. Landslide materials move downslope until they reach a stable position. Whereas some landslides travel only partway downslope (Fig. 1b), others

reach drainages and mobilize into debris flows that can severely impact roads and infrastructure. Non-channelized runout zones exist downslope of landslide source zones, where the source zone is not adjacent to the channel or in open-slope topographies (Fig. 1a). In open-slope topographies, no channels are present and landslides travel downslope without entering a drainage or topographic depression (Geotechnical Engineering Office, 2012). Where channels are present, highly mobile debris flows will travel into the channel, potentially grow in volume, and flow long distances downstream (Fig. 1c). Our empirical runout models

allow topography to control the spatial distribution and extent of potential landslide runout and debris-flow inundation zones. Our approach simulates patterns consistent with observations from Hurricane Maria when applied over a topographically diverse area, including the full extent of three municipalities: Lares, Utuado and Naranjito. Our USGS software package, Grfin Tools (Cronkite-Ratcliff et al., in review; Reid et al., in review) implements these methods and enables runout assessment over large regions without the computational effort required by physics-based models.

Public domain. CC0 1.0.

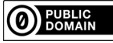

Natural Hazards
and Earth System
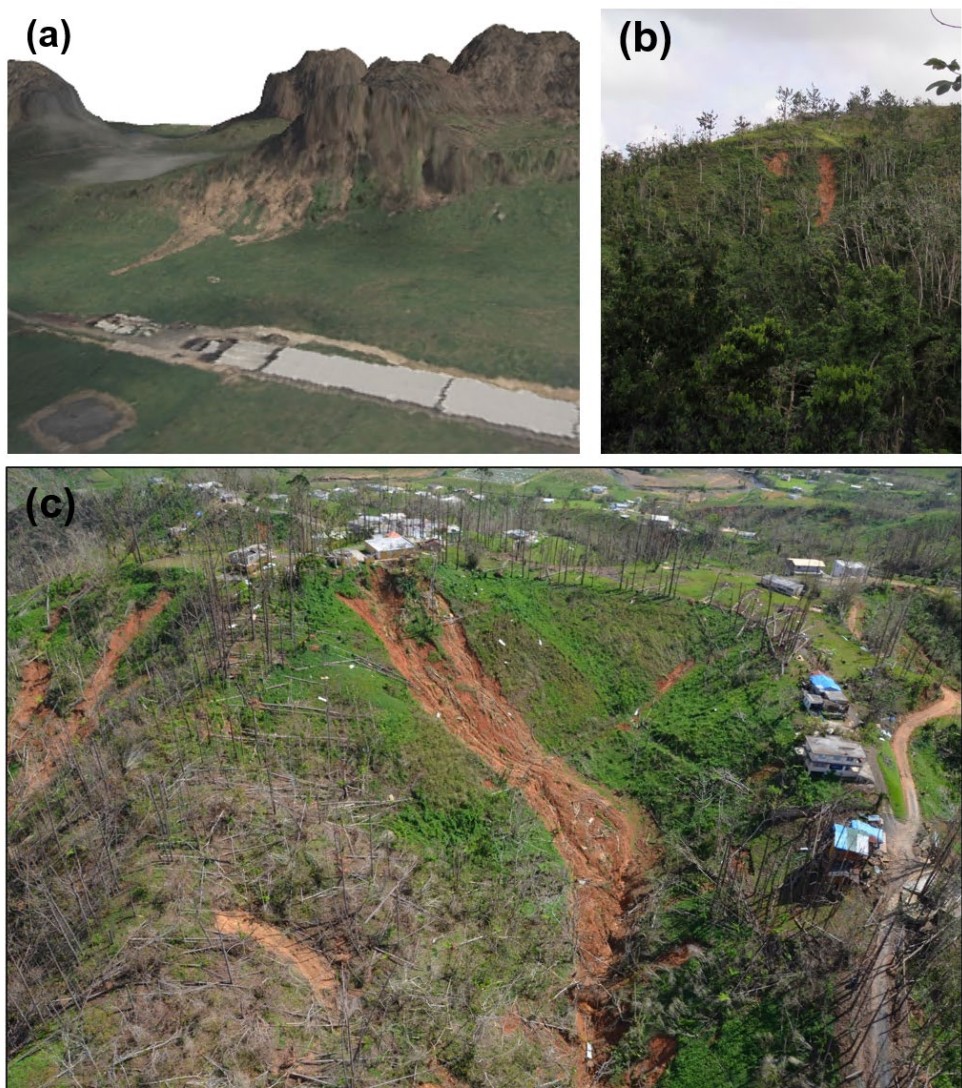


**Figure 1. Photographs showing Hurricane Maria landslides with varied levels of mobility a) Orthophoto (Quantum Spatial 2017) draped on DEM (U.S. Geological Survey 2020), showing multiple landslides in non-channelized, open-slope topography of northern Utuado, adjacent to a cone karst topography. b) Photograph of two moderate-mobility, shallow landslides, in Utuado, Puerto Rico, that mobilized and travelled partway downslope but insufficient distance to reach a channel. c) Photograph of higher-mobility debris flow that initiated**
**from multiple shallow landslides and entered channelized topography in the lower half of photo, in the Ciales municipality, Puerto Rico. Photograph in c by Jason Kean (USGS).**

Public domain. CC0 1.0.





## 2 Conceptual framework

### 2.1 Zones of mobility

Our conceptual framework uses three zones of mobility within the landscape: 1) source zones, 2) non-channelized runout zones, and 3) channelized runout zones (debris-flow inundation zones). This framework provides the foundation to investigate two interrelated aspects of the Hurricane Maria landslides: 1) a topographic analysis of published landslide inventories and 2) a modeling approach to assess susceptibility to non-channelized and channelized runout in Puerto Rico. Results of the topographic analysis inform our selection of model parameters for candidate susceptibility scenarios, with the overall objective to select two final scenarios for regional susceptibility maps.

### 2.2 Topographic analysis

Our topographic analysis guided the selection of input parameters for runout modeling and provided a complementary analysis associated with zones of mobility, enabling us to gain perspectives on the landslide-affected areas and relative contributions of each zone of mobility. Published landslide inventories of Hurricane Maria landslides (Bessette-Kirton et al., 2019b; Baxstrom et al., 2021a, 2021b; Einbund et al., 2021a, 2021b) provided the location of landslide-affected areas for our topographic analysis,
whereby we analyzed the percentage of area affected by each zone of mobility for Hurricane Maria landslides and extracted slope characteristics within each zone. We also assessed correlations between study area slope and the slope of source areas and non-channelized runout, as trends could influence whether different parameters are needed for runout modeling as a function of geologic or topographic variability. In addition, we identified a subset of the mapped landslides, representative of the most mobile channelized debris flows. Typical characteristics of the inundation zones associated with the most mobile debris flows
allowed us to define parameters for potential zones of debris-flow growth in our debris-flow inundation modeling. These inundation zones provided an important component for assessment of the predictive success of our inundation methods.

### 2.2 Linked-model approach

We developed a linked-model approach that combines landslide source with two methods to identify areas susceptible to landslide runout and debris-flow inundation. Here, the "link" is joined independently between potential landslide source areas
and each runout method (Fig. 2).

The runout methods differ based on the relative mobility and topographic setting of landslides. For moderate mobility landslides and/or non-channelized runout zones, we define potential runout zones by minimum angle of reach (arctan(height/length)) from the landslide source. This approach provides a methodology to 1) estimate runout in open-slope topographies where channels are not present, and 2) provide a transition from upslope landslide source zones to channels.

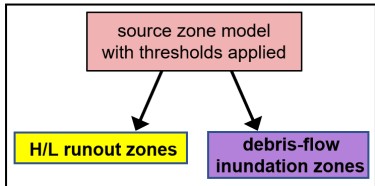


**Figure 2. Schematic of linked-model approach.**

Public domain. CC0 1.0.





Although other types of landslides may flow, the definition of debris flow provided by Hungr et al. (2014) is well aligned with our modeling approach for inundation zones: "Very rapid to extremely rapid surging flow of saturated debris in a steep channel. Strong entrainment of material and water from the flow path". These debris flows can increase in volume as they travel, due to a combination of processes, including entrainment of bed sediment (e.g., Hungr et al., 1984; Takahashi, 1991; Iverson et al., 2011), coalescence of landslides (e.g., Coe et al., 2021), and stream bank collapse (Johnson, 1970). For channelized debris flows, we identify potential inundation zones using empirical volume-area relations (Griswold and Iverson, 2008) in concert with empirical debris-flow growth factors (Reid et al., 2016). Our growth factors integrate growth over a drainage basin and are defined as a function of upstream contributing area susceptible to shallow landslides. This approach determines the spatial distribution and volumes of runout material contributing to debris-flow inundation zones.

Our linked-model approach 1) provides three zones of hazard (landslide source, non-channelized runout, and channelized runout) related to landslide mobility, 2) uses angle of reach to identify potential non-channelized runout zones, 3) incorporates debris-flow growth for channelized debris flows, 4) estimates debris-flow volumes as a function of contributing area susceptible to landslides, and 5) applies volume-area relations to estimate corresponding areas of debris-flow inundation. Combined, this approach provides a GIS-based method, applicable over diverse terrains of varied susceptibility to debris flows. Our USGS software package, Grfin (gr=growth + f=flow + in=inundation; pronounced griffin) Tools (Cronkite-Ratcliff et al., in review; Reid et al., in review) implements this approach and enables runout assessment over large regions without the computational effort typically required by physics-based models. We used our linked-model approach to create regional susceptibility maps of landslide initiation and runout in the three municipalities of Lares, Utuado and Naranjito, Puerto Rico.

## 3 Study areas

Steep mountainous terrain, high mean annual rainfall, and frequent intense storms in Puerto Rico contribute to the frequent occurrence of landslides, resulting in extensive property damage and loss of life (e.g., Larsen and Torres-Sanchez, 1998). Rainfall-triggered landslides are the most common type of landslide, occurring throughout the central mountains and foothills of the island, as frequently as 1 to 2 times per year (Larsen and Simon, 1993).

On 20 September 2017, Hurricane Maria produced rainfall amounts greater than any other hurricane or tropical storm in Puerto Rico since 1956; within a 48-hour period, at least 250 mm of rain fell across Puerto Rico's mountainous terrain (e.g., Bessette-Kirton et al., 2019a) with as much as 1029 mm of precipitation recorded in the southeastern part of the island (Keellings and Hernández Ayala, 2019). Hurricane Maria triggered more than 70,000 landslides in Puerto Rico (Hughes et al., 2019). Our work builds on published data sources related to the widespread landsliding that occurred during Hurricane Maria.

### 3.1 Data sources and related work

#### 3.1.1 Topographic base

We used high-resolution pre- and post- Hurricane Maria lidar-derived DEMs to construct a channel network and determine flow directions for our runout modeling. A pre-Maria, 1 m resolution DEM, acquired between January 2016 and March 2017 (U.S. Geological Survey, 2018) was representative of the topography at the time of Hurricane Maria. This pre-Maria DEM was used for extraction of topographic characteristics and assessment of model predictive success. A 0.5 m resolution, post-Maria lidar-derived DEM (U.S. Geological Survey, 2020a,b,c) was resampled to 1 m and used to create regional susceptibility maps of landslide initiation and runout.

Public domain. CC0 1.0.


### 3.1.2 Mapped landslide inventories

Published landslide inventories (Bessette-Kirton et al., 2019b; Baxstrom et al., 2021a, 2021b; Einbund et al., 2021a, 2021b)

provided detailed mapping of landslide-affected areas from Hurricane Maria, including 2919 locations of landslide headscarp points, travel distance lines, landslide-affected areas, and source-area-only locations. Lengths were measured from the top of the headscarp to the farthest extent of visible landslide deposits (Bessette-Kirton et al., 2020). Hurricane Maria source-area locations were determined from pre- and post-event lidar-derived DEM differences (2016 to 2018) (U.S. Geological Survey, 2018, 2020a,b,c).

The inventories encompassed nine study areas, within three municipalities, over four distinctive geologic terranes — defined as groups of geologic formations, based on lithologic rock type, depositional environment, and/or age (Bawiec, 1998) (Table 1). Based on municipality, we applied naming conventions to identify the study areas. The Utuado municipality includes four ~2.5 km² study areas (U1, U2, U3, U4, Fig. 3) in a granitoid terrane (Ku) consisting of the Utuado batholith; these four areas are further distinguished by two geomorphic terrains (Einbund et al., 2021b): 1) escarpments (U1, U2) having highly dissected

areas with predominantly steep topography and high drainage density and 2) upland terrain (U3, U4) consisting of dissected, low-relief plateaus (Monroe, 1980) with lower drainage density, relative to escarpments. Study areas U5, L1, L2, L3, and N (Fig. 3) do not have a distinctive plateau expression and do not contain a single unique geomorphic terrain. Northern Utuado includes the largest (~30 km²) study area (U5), with low landslide density, located in non-limey sedimentary units (Baxstrom et al., 2021b) adjacent to cone-karst topography, where conical, steep-sided hills of the Lares Limestone, named mogotes, rise to

heights up to 100 m (Monroe, 1976). The Lares municipality includes three ~3.5 km² study areas (L1, L2, L3) (Einbund et al., 2021a), located in Tertiary-Cretaceous marine volcaniclastics, consisting mainly of breccia, tuff, sandstone, and siltstone (Tka/Tkal). Naranjito municipality contains one ~2.5 km² study area (N) (Bessette-Kirton et al., 2019b; Baxstrom et al., 2021a), located in Cretaceous marine volcaniclastics, consisting mainly of basaltic breccia, sandstone, and siltstone (Kln).

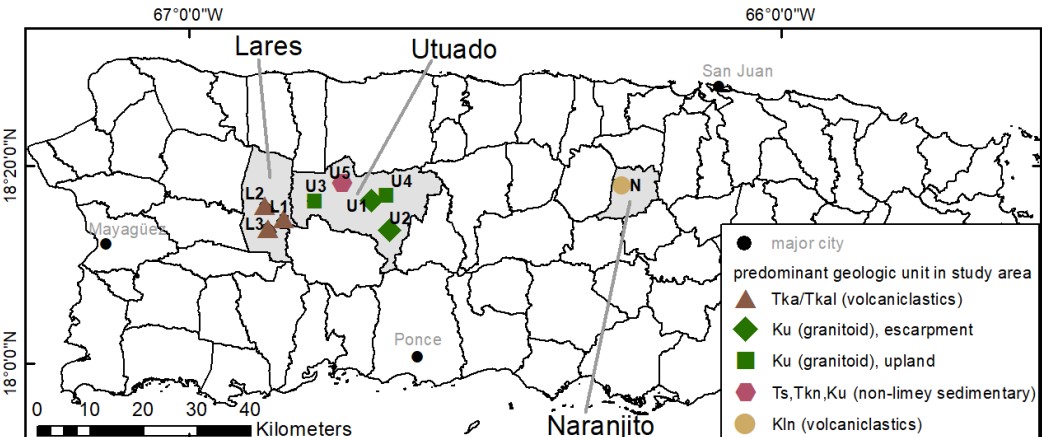

**Figure 3. Map of Puerto Rico, showing locations of nine study areas with detailed landslide mapping of 2919 landslides in the Lares, Utuado, and Naranjito municipalities (Bessette-Kirton et al., 2019b; Baxstrom et al., 2021a, 2021b; Einbund et al., 2021a, 2021b). Study-area name is indicated by the first letter of the municipality, followed by a numeral.**

**Table 2. Study area names, geologic terrane, predominant geologic units (Bawiec, 1998), and geomorphic terrains (escarpment or upland) for nine areas with mapped landslide-affected areas (Bessette-Kirton et al., 2019b; Baxstrom et al., 2021a, 2021b; Einbund et al., 2021a,**

**2021b). Color and symbol combinations indicate geologic terrane; within the granodiorite. Two unique symbols are used to distinguish escarpment (green diamonds) versus upland (green squares) geomorphic terrains.**

Public domain. CC0 1.0.
| symbol | study area name | geologic terrane | predominant geologic units | geomorphic terrain |
|---|---|---|---|---|
| ◆ | U1 | granitoid | Ku (Utuado batholith) | escarpment |
| ◆ | U2 | granitoid | Ku | escarpment |
| ■ | U3 | granitoid | Ku | upland |
| ■ | U4 | granitoid | Ku | upland |
| ⬡ | U5 | non-limey sedimentary | Ts (San Sebastian Formation), Tkn (Naranjito Formation) | not distinguished |
| ▲ | L1 | marine volcaniclastic | Tka (Anon Formation), Tkal (Anon and Lago Garzas Formations) | not distinguished |
| ▲ | L2 | marine volcaniclastic | Tka, Tkal | not distinguished |
| ▲ | L3 | marine volcaniclastic | Tka, Tkal | not distinguished |
| ● | N | marine volcaniclastic | Kln (Los Negros Formation) | not distinguished |

### 3.1.3 Landslide types

Landslides triggered by Hurricane Maria included slumps, debris flows, rockfalls, and other slope failures (Hughes et al., 2019).

Most the landslides were shallow debris slides and many of these mobilized and/or coalesced into channelized debris flows (Bessette-Kirton et al., 2020; Coe et al., 2021). Figure 4 shows the variety of landslide styles associated with channelized (Fig. 4b) and non-channelized (Fig. 4a) topography. In adjacent drainages, landslide density and the mobility of these landslides can be widely varied (Fig. 4b).

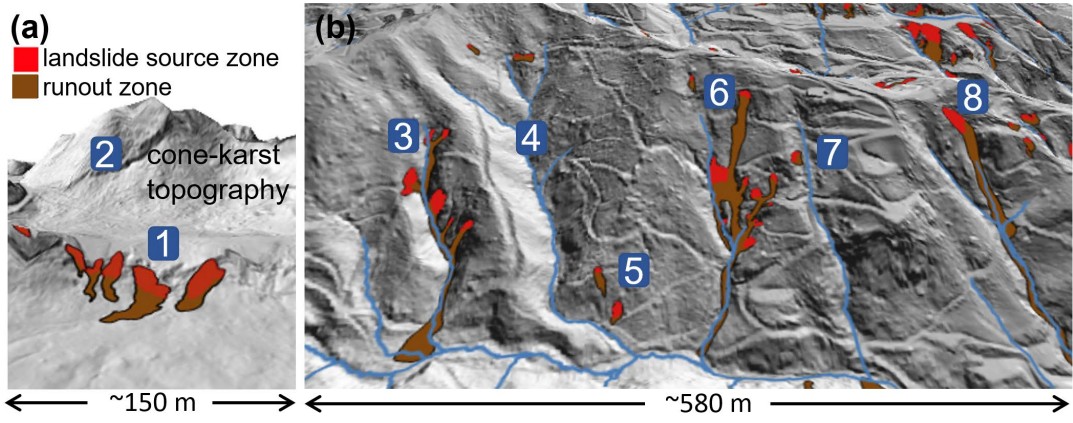


**Figure 4. Perspective views showing topographic features, mapped landslide source areas, and runout in small sections of two study areas. a) Landslides on non-channelized open-slope terrain (1) in Northern Utuado (study area U5) (Baxstrom et al., 2021b), adjacent to cone-karst topography (2). b) Study area N (Bessette-Kirton et al., 2019b; Baxstrom et al., 2021a), showing basins with varying landslide density and landslide types: 3. basin affected by landslides coalescing into debris flows, 4. unaffected basin, 5. low mobility**

**landslides on cut-slopes adjacent to roads, 6. landslide on hillslope coalescing with multiple landslides closer to or located in drainage, 7. basin with one-small, low mobility landslide, and 8. a single landslide near the top of the hillslope that mobilized as a channelized debris flow. Approximate location of a at center of image is 18° 18′ 10″ N, 66° 49′ 10″; b is located at 18° 18 ′0″ N, 66° 16′ 0″ W.**

Public domain. CC0 1.0.

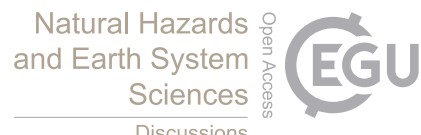

### 3.1.4 Potential landslide source areas

To estimate potential landslide source areas in our linked-model approach, we used areas identified from the combination of soil-depth estimations, pore-water pressures, and slope stability analysis. The areas susceptible to shallow landslides during prolonged, intense rainfall were defined by factor of safety thresholds for high and very high susceptibility scenarios. Details of the source area modeling are provided in Baum et al. (2024).

### 3.1.5 Debris-flow growth and volumes

For our debris-flow inundation modeling, we used published estimates of debris-flow growth factors and volumes. These estimates (Table 2) were based on lidar-derived DEM differencing in four drainage basins affected by long-runout debris flows from Hurricane Maria (Coe et al., 2021). Growth factors based on upslope contributing areas are shown as a function of both i) full contributing area and ii) area susceptible to landslides, approximated as slopes greater than 30° (Coe et al., 2021), where i is applicable to basins of similar susceptibility and ii is applicable to regions with spatially variable landslide susceptibility patterns.

Field measurements of stream slopes for several Hurricane Maria debris flows were measured using a laser rangefinder with inclinometer (Coe et al., 2021). These measurements showed that growth transitioned to deposition at a stream slope between 3 to 8°, providing constraints on debris-flow growth zones for our modeling.

**Table 2. Range of values for debris-flow growth factors and total volumes from Hurricane Maria debris flows (Coe et al., 2021).**

|  | **range of values** |
|---|---|
| *i*) area-based growth factors, full contributing area ($c_1$) | $0.01{-}0.13 \ \mathrm{m^3 \, m^{-2}}$ |
| *ii*) area-based growth factors ($c_1$), calculated as the percentage of area with slopes > 30° | $0.02{-}0.21 \ \mathrm{m^3 \, m^{-2}}$ |
| *iv*) total volumes (*V*) | $840{-}12{,}770 \ \mathrm{m^3}$ |

### 4 Methods

Figure 5 shows a flowchart illustrating the methods for our topographic analysis and linked-model approach, where blue boxes and one red box indicate existing data sources. The left side the flow chart (gray) shows our workflow for topographic analysis and the right side (tan) shows the steps for our linked-model approach. The three components of the linked-model are shown: 1) source (red), 2) non-channelized runout (yellow), and 3) debris-flow inundation (purple).

Public domain. CC0 1.0.
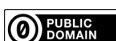

**Figure 5. Flowchart of our topographic analysis and linked-model approach. Blue boxes (and one red box) indicate published data sources. Red, yellow, and purple boxes indicate the three components of our linked-model approach.**

## 4.1 Channel network delineation

Delineation of a channel network derived from the 1 m DEMs was essential for derivation of flow direction and flow accumulation for runout modeling, as well as the distinction of non-channelized versus channelized runout zones. Roads are an

Public domain. CC0 1.0.
inherent problem for channel detection algorithms that use high-resolution topographic data, as roads often obscure the topography of natural channels. In Puerto Rico, large municipal roads, along with small agricultural and private roads (Ramos-Scharrón et al., 2021), led to significant disruption of the flow directions derived from the DEMs. Whereas some debris flows from Hurricane Maria were diverted by these agricultural roads, the majority bypassed roads and continued down natural channels (Bessette-Kirton et al., 2019a). Accurately modeling debris-flow inundation required defining downstream channel

networks that were continuous across laterally intersecting road networks.

     Our automated methods applied two strategies to eliminate road-network artifacts from the lidar-derived DEMs and construct channel networks representative of natural channels from the lidar-derived DEM: 1) identification of the location of channel initiation using a curvature-based flow accumulation threshold, and 2) spectral filtering of the DEMs to remove road artifacts downstream of channel initiation.

### 4.1.1 Curvature-based method to identify channel initiation


Our implementation of curvature-based network delineation was inspired by Tarboton and Ames (2001) and used a flow accumulation threshold including only topographic concavities (hollows) that are representative locations of channel initiation. To detect channel initiation points for our drainage network, we used concave planform curvature to identify areas representative of topographic hollows.

Steps in our method to identify locations of channel initiation included: 1) using a local mean to smooth the DEM, 2) calculating planform curvature, 3) applying a curvature threshold to identify concavities in the topography, 4) eliminating small, isolated concavities, 5) calculating the contributing area of remaining concavities, and 6) applying an area threshold using only the contributing area of concave topography. We used threshold values based on topographic scaling factors and published values (Pelletier, 2013; Mudd et al., 2019). The location of channel initiation was assigned where the contributing concave area,

defined as a planform curvature $< 0.02$ m$^{-1}$, was larger than 500 m$^2$.

### 4.1.2 Bandpass DEM to derive flow directions

In areas downstream of channel initiation, the channel network and associated flow directions from the lidar-derived DEMs were sometimes diverted at road intersections. We employed a spectral filtering approach to remove small roads from the topography. Using SpecFiltTools software (Perron et al., 2008), we applied a Gaussian bandpass filter to remove topographic features at the

wavelength of small agricultural roads (~10 m) while retaining both the smaller and larger wavelengths that constitute the undisturbed topography. The combination of our two strategies provided an automated method to remove roads from channel networks over diverse geomorphic and geologic terranes.

### 4.2 Topographic analysis of mobility zones

Using the published datasets of source areas and landslide-affected areas (Bessette-Kirton et al., 2019b; Baxstrom et al., 2021a,

2021b; Einbund et al., 2021a, 2021b), combined with our channel network, we divided the mapped landslide-affected areas in the nine study areas into the three zones of mobility. The published mapped source areas provided the first zone of mobility. Using ArcGIS® spatial analyst tools by Esri, we divided the remaining area into non-channelized and channelized runout. Runout zones from small landslides (length $< 20$ m) with a small percentage ($< 20\%$) of area adjacent to the drainage or distanced greater than 3 m from the delineated drainage network were assigned as non-channelized runout and the remaining

area was designated as channelized runout.

Public domain. CC0 1.0.

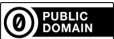



Within the three mobility zones, we: 1) evaluated the percentage of each study area affected by each zone, 2) analyzed summary statistics to compare the distribution of topographic slopes in the three mobility zones, and 3) compared variability between study areas. Summary statistics were analyzed based on every raster cell within a mobility zone.

To evaluate the percentage of each study area affected by landslide source zones, we considered: 1) percentage of full study area, and 2) percentage of each study area susceptible to landslides, approximated by steep slopes greater than 30° (Coe et al., 2021). This approximation of susceptible areas was consistent with field observations of Baum et al. (2018), and provides a justifiable criterion to calculate normalized values, whereby, the percentage of area affected would be equal across all study areas if all other contributing factors were equal.

Summary statistics, for each mobility zone in each study area, included percentiles and the Fisher-Pearson coefficient of skewness ($G_1$) (Zwillinger and Kokoska, 2000), a measure of the asymmetry of a statistical distribution, where $-0.5 < G_1 < 0.5$ indicates the data are approximately symmetric; $-1 < G_1 < -0.5$ indicates the data are moderately left-skewed; $0.5 < G_1 < 1$ indicates the data are moderately right-skewed; $G_1 < -1$ indicates the data are highly left-skewed; $G_1 > 1$ indicates the data are highly right-skewed (Brown, 2022).

### 4.2.1 Hurricane Maria's most mobile (MMM) landslides

Next, we developed criteria to extract a subset of mapped landslide-affected areas representative of Hurricane Maria's most mobile channelized debris flows (MMM). The identification of MMM provided a dataset for assessment of predictive success of our debris-flow inundation modeling. Characterization of the runout zones associated with the MMM debris flows guided the parameterization of debris-flow growth zones.

The primary criterion to identify MMM was a high (> 40%) percentage of runout area located in close proximity (< 5 m) to a designated channel. This criterion identified channelized debris flows as well as some less mobile, non-channelized runout zones located close to the channel. To eliminate these non-channelized runout zones, we applied an additional criterion, runout length >100 m. We used runout lengths available in the published landslide inventories. Given that post-event evidence did not provide information to distinguish specific contributions from sources upstream of tributary junctions, coalescing runout zones were grouped together; one runout zone may represent the path of a single debris flow or many coalescing debris flows. 265     We applied these two criteria to extract MMM for the nine study areas (Brien et al., in press).

To evaluate representative characteristics of channelized debris-flow runout zones and define constraints on debris-flow growth zones for inundation modeling, we compiled percentile statistics from the runout zones associated with MMM. These zones may have included stream reaches of growth, transport and/or deposition. We were not able to identify reaches with only debris-flow growth given the available information, therefore the values extracted represent extremes of reasonable values to 270     constrain debris-flow growth zones. Due to the small sample size for upland terrains for this analysis, we grouped the study areas by geologic terrane.

We computed percentile statistics within the runout zones, where the maximum value of stream order provided the most useful statistic to constrain characteristics of growth zones and other variables described below were characterized by means. For each runout zone, we determined the maximum Strahler stream order, mean stream slope calculated over a horizontal distance of 275     50 m, mean planform curvature from a smoothed DEM, and mean percentage of contributing area susceptible to shallow landslides ($P_{src}$), where:

$$P_{src} = 100 \left( \frac{A_{src}}{A} \right) \qquad\qquad (1)$$

Public domain. CC0 1.0.

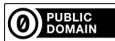



$A_{src}$ is total contributing area susceptible to shallow landslides, estimated from source-zone modeling (Baum et al., 2024) and $A$ is total contributing area. Contributing areas and corresponding values for $P_{src}$ were calculated for each raster cell within a runout

zone using a single direction *D-8* flow model (Tarboton et al., 2015).

### 4.3 Linked-model approach

### 4.3.1 Landslide initiation zones

Potential source areas for our runout methods can be obtained from any empirical- (e.g., Furbish and Rice, 1983; Larsen and Parks, 1998; Lepore et al., 2012; Hughes and Schulz, 2020; Merghadi et al., 2020) or physics-based (e.g., Montgomery and

Dietrich, 1994; Pack et al., 1999; Baum et al., 2008; Mergili et al., 2014; Reid et al., 2015) landslide susceptibility methods. For Puerto Rico, we used slope stability analysis results to identify potential source areas. Baum et al. (2024) defined factor of safety thresholds needed to capture 0.75 (high susceptibility) and 0.90 (moderate susceptibility) true positive rate for observed headscarp points (from Hughes et al., 2019) of landslides triggered by Hurricane Maria. For our non-channelized runout zones, discussed below, we used the combined high and moderate susceptibility potential source areas. The high-susceptibility areas

were used as the upslope contributing source area for debris-flow inundation scenarios.

### 4.3.2 Delineation of non-channelized runout zones

We identified runout zones for moderate mobility landslides using *H/L* runout zones delineated with the avalanche runout tool from TauDEM toolbox (Tarboton et al., 2015). This tool uses a *D-infinity* method to determine flow directions along a flow path (Tarboton, 1997). Length downslope of potential source areas is limited by a threshold angle equivalent to arctan (*H/L*) or the

angle of reach (α) (e.g., Scheidegger, 1973; Hsu, 1975; Nicoletti and Sorriso-Valvo, 1991; Corominas, 1996; Iverson et al., 2015; Legros, 2002; Wallace et al., 2022), where $H$ is defined as the vertical drop, and $L$ is the horizontal projection of distance. On a hillslope or in a DEM, the flow path for measurement of the horizontal length, $L$ may follow a winding pathway downslope and down-channel. In locations where runout enters a channel, the runout zones defined by *H/L* are not able to delineate width of inundation. This limitation of *H/L* runout estimates is addressed by our application of debris-flow inundation modeling in

channelized topography.

### 4.3.3 Delineation of channelized debris-flow inundation zones

For high mobility, channelized debris flows that grow as they travel, we identified zones of potential debris-flow growth, calculated debris-flow volumes using debris-flow growth factors (Reid et al., 2016) and identified areas susceptible to inundation with Grfin Tools debris-flow inundation model (Cronkite-Ratcliff et al., in review; Reid et al., in review). The Grfin Tools

implementation eliminates spiky artifacts that can be present in results using other empirical debris-flow inundation models such as Laharz (Schilling, 2014) or DFLOWZ (Berti and Simoni, 2014).

Our modeling used a semiempirical approach relating volume with cross-sectional and planimetric area (Iverson et al., 1998), allowing us to estimate inundation area from debris flows. This approach uses power-law relations for debris-flow inundation (Griswold and Iverson, 2008) combined with empirical growth factors (Reid et al., 2016; Coe et al., 2021).

Planimetric and cross-sectional inundation area estimations are calculated from two statistically derived equations, based on a worldwide database of debris-flow measurements from diverse data sources and geographic locations, ranging in volume from 10 to $10^6$ m³ (Griswold and Iverson, 2008):

$$A = 0.1\,V^{2/3} \tag{2}$$


$$B = 20\, V^{2/3} \tag{3}$$


where $A$ is cross-sectional area, $B$ is planimetric area, and $V$ is debris-flow volume. Previous studies of non-post-wildfire debris flows yield similar coefficients for these relations, where the cross-sectional area coefficient ranged from 0.07 to 0.1 and the planimetric area coefficient ranged from 17 to 20 (Griswold and Iverson, 2008; Berti and Simoni, 2014). The estimated cross-sectional area and planimetric areas are applied to a DEM to define areas susceptible to channelized debris flows.

320        Debris-flow volume is of foremost importance for this approach and for inundation modeling in general. Previous studies indicate debris-flow inundation patterns and flow depth estimates may be more sensitive to flow volume than flow properties (Barnhart et al., 2021). We compute volume as a function of upslope contributing area susceptible to shallow landslides at locations in the digitally derived channel network where debris-flow growth is likely to occur (growth zones):

$$V = \begin{cases} c_1\, A_{src} & if\ \ c_1\, A_{src} < V_{max} \\ V_{max} & if\ \ c_1\, A_{src} \ge V_{max} \end{cases} \tag{4}$$

where $V$ is debris-flow volume, $c_1$ (units of $L^3\,L^{-2}$) is an empirically derived growth factor (Reid et al. 2016), $A_{src}$ is potential upslope contributing source area, and $V_{max}$ is maximum volume. Volumes are ultimately constrained by $V_{max}$, based on volumes estimates from Hurricane Maria. Using volumes from Eq. (4), cross-sectional and planimetric inundation areas can be derived using Eqs. (2, 3).

        For Puerto Rico, volumes calculated as a function of areas susceptible to debris flows ($A_{src}$), based on Baum et al.
(2024), allow us to apply these empirical relations over large regions with varied geologic terranes and geomorphic terrains where landslide susceptibility is spatially variable. Equation 4 provides volumes regulated by $c_1$ and $A_{src}$. Basins with minimal susceptible area result in smaller volumes and basins of high susceptibility produce larger volumes, limited by $V_{max}$. For areas with no susceptible contributing source area, debris-flow volumes are nil, and no inundation will be estimated. We use the term "self-regulating" volumes to describe volumes estimated by Eq. (4).

**4.4 Parameters and assessment of linked-model approach**

**4.4.1 Selection of height/length ($H/L$) values for regional susceptibility maps**

To select $H/L$ values for regional susceptibility maps, we considered a range of previously published $H/L$ values in Puerto Rico as well as global datasets. In Puerto Rico, Bessette-Kirton et al. (2020) calculated median $H/L$ values for 1035 landslides from Hurricane Maria as 0.68 ($\alpha = 34°$) and coalescing landslides as 0.52 ($\alpha = 27°$), with median lengths ($L$) of 17.5 m and 25.2 m,
respectively. These values represent typical Maria-induced landslides with relatively short travel distance, as reflected by the median lengths. The most mobile landslides had $H/L$ values less than 0.25 ($\alpha = 14°$, Bessette-Kirton et al., 2020).

        Other published $H/L$ data for landslides in the volume range from Hurricane Maria include regression equations quantifying the relation between the angle of reach ($\alpha$) and landslide volume; for all landslide types combined, $\log (H/L) = -0.085\log V - 0.047$ which yields $H/L$ values of 0.61 ($\alpha = 31°$) to 0.41 ($\alpha = 22°$) for landslide volumes of 100 to 10,000 m$^3$
respectively (Corominas, 1996). Data from flume experiments yields $L/H = {\sim}2$ (equivalent to $\alpha = {\sim}27°$) for unconfined runout but greater than 2 for channelized runout, for volumes of ${\sim}10$ m$^3$ (Iverson, 1997). Given the wide range of published values, we used mapped landslide source areas to assess the change in area affected by $H/L$ runout over a wide range of $\alpha$ values: 10, 15, 20, 25 and 30°.

Public domain. CC0 1.0.

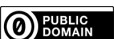


#### 4.4.2 Debris-flow growth zones and volumes

Debris-flow volumes (Eq. (4)), calculated as a function of upslope contributing area susceptible to shallow landslides, were computed where debris-flow growth is likely to occur. Debris-flow growth zones were defined by a combination of parameters, including stream slope, stream order, planform curvature, and $P_{src}$ (Eq. (1)). The rate of growth was controlled by $c_1$, limited by $V_{max}$ (Eq. (4)).

We considered eight debris-flow inundation scenarios (Fig. 6), constrained by 1) the minimum stream slope where
growth transitioned to deposition (3°) (Coe et al., 2021), 2) MMM statistics for stream order, stream slope, curvature, and $P_{src}$, and 3) published debris-flow volumes and growth factors ($c_1$) (Coe et al., 2021). All scenarios excluded channel sections with planform curvature < 0.02 m$^{-1}$ and $P_{src}$ < 20%, sections unlikely to produce debris-flow growth based on 75–90% of MMM. Unrealistically short (< 4 m) stream segments of channel identified as potential growth zones were also excluded. The final selection of two scenarios for region-wide susceptibility maps was based on evaluation predictive success the inundation results
produced from these eight scenarios.

Columns in the matrix of scenarios (Fig. 6) identify debris-flow growth zone scenarios (A, B, C, D, E). Rows identify associated parameters for debris-flow volumes, including maximum volumes of 1000, 3000, 5000, and 10,000 m$^3$. Each scenario is assigned an identifier, such as A-1k, based on a combination of the associated letter for growth zone scenario (A) and assigned maximum volume (1k).




**Figure 6. Matrix of eight debris-flow inundation scenarios considered in three municipalities. Red outlines highlight two final scenarios selected.**

#### 4.4.3 Assessment of predictive success for *H/L*

The assessment of predictive success for *H/L* zones was not easily quantifiable given that the landslide inventories were focused on areas with the highest landslide densities. In these high density areas, steep (> 30°) slopes led directly to channelized zones and non-channelized runout typically reflected the local topographic slope. In open-slope terrains, a limited number of mapped landslides was represented in the available inventories. We used source areas from the published landslide inventories and considered the spatial patterns in affected areas (non-channelized versus channelized) and increase in estimated runout area for a
range of α values between 10 and 30°.

Public domain. CC0 1.0.

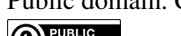



### 4.4.4 Assessment of predictive success of debris-flow inundation

In a back-analysis mode commonly used in evaluation of debris-flow runout (McDougall, 2017), we evaluated the success of the eight debris-flow inundation scenarios to predict the presence and extent of 124 debris-flow inundation zones from MMM. Our assessment used contingency table statistics and standard receiver-operator characteristic (ROC) analysis (Powers, 2011) to

analyze the predictive success of our results. ROC analysis is based on statistics computed from a binary contingency table, whereby four categories of predictive success are identified: 1) true positive (*TP*) indicates successful prediction of an area susceptible to landslide runout, 2) false positive (*FP*) indicates false prediction of susceptible area, 3) false negative (*FN*) indicates a susceptible area was not identified, and 4) true negative (*TN*) indicates successful prediction of a stable area. We considered three measures of predictive success: 1) true positive rate, $TPR = TP/(TP+FN)$; 2) false positive rate, $FPR = $

$FP/(FP+TN)$; and 3) positive likelihood ratio, $PLR = TPR/FPR$.

To select two scenarios for regional susceptibility maps, our ROC analysis used the intersection of 1) inundation zones from Hurricane Maria (MMM) within 5 m of the channel thalweg, and 2) the area encompassed by all inundation scenarios combined. This method evaluated a combination of inundation width and length, but was deleteriously influenced by minor georeferencing discrepancies between the mapped landslides and lidar-derived DEM, as well as underestimation of runout

length, where the terminus of debris-flow deposits could not be discerned in the aerial photographs due to uncertainty in debris-flow extent for flows entering drainages where floodwaters reworked deposits (Bessette-Kirton et al., 2019b; Baxstrom et al., 2021a, 2021b; Einbund et al., 2021a, 2021b). For the two selected scenarios, we also assessed TPR to determine success in identification of MMM inundation zones, a method that does not consider runout width or length.

## 5 Results of topographic analysis

### 5.1 Zones of mobility

Figure 7 shows landslide-affected areas (Einbund et al., 2021b), divided into the three zones of mobility: source areas are shown in red, non-channelized runout zones are shown in yellow, and channelized runout zones are shown in purple.

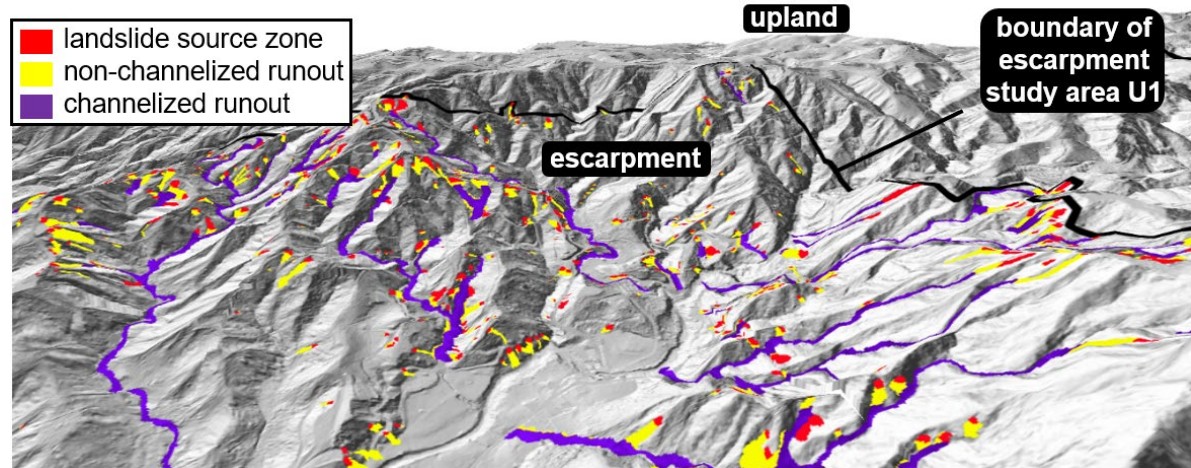

**Figure 7.  Perspective view showing landslide-affected areas (Einbund et al., 2021b) for study area U1, on an escarpment in the Utuado**
**municipality, divided into three zones of mobility: 1) source zones (red areas), 2) non-channelized runout (yellow areas), and 3) channelized runout (purple areas). For reference, a dissected upland terrain, representative of topography in U3 and U4, is visible in the background. Approximate location of at center of image is 18° 16′ 40″ N, 66° 41′ 20″ W.**

Public domain. CC0 1.0.



**5.2 Area affected by zones of mobility**

We assessed the percentage of area affected by each mobility zone in the nine study areas (Table 3). The percentage of the study
areas with mapped landslide source zones are shown both as the percentage of study area susceptible to landslides, approximated
as slopes > 30° (column 3), and the percentage of the entire study area (column 4). Although all study areas encompass some
steep topography susceptible to landslides, escarpments (U1, U2) have a majority of the study area susceptible, whereas
dissected uplands (U3, U4) have lower relief and a smaller percentage of the study area in steep ground (Table 3, column 2 and
Fig. 8). Note that the other five study areas (U5, L1, L2, L3, N) contained mixed topography and were not specifically separated
into escarpment versus upland terrains.

**Table 3. Percentage of nine study areas susceptible to landslides (column 2) and affected by landslides during Hurricane Maria (column 7) divided into three zones of mobility: source zone (columns 3 and 4), non-channelized runout (column 5), and channelized runout (column 6). Column 7 shows the sum of these three zones. The percentage of the study areas with landslide source zones are shown as the percentage study area susceptible to landslides (column 3) and the percentage of the entire study area (column 4). Study**
**areas are listed in order of increasing percentage of area susceptible to landslides (column 2).**

| symbol | study area | size (km²) (1) | % study area susceptible to landslides (steep slopes > 30°) | | % study area in landslide-affected areas | | | |
|---|---|---|---|---|---|---|---|---|
| | | | % area with steep slopes (2) | % steep areas with landslide source (3) | % area with source zone (4) | % area with non-channelized runout (5) | % area with channelized runout (6) | % area affected by landslides (total) (7) |
| 🔴 | U5 | 28.5 | 30.1% | 0.4% | 0.1% | 0.1% | 0.1% | 0.3% |
| 🟩 | U3 | 2.5 | 32.6% | 2.4% | 0.8% | 0.9% | 0.8% | 2.6% |
| 🟩 | U4 | 2.5 | 33.0% | 2.1% | 0.7% | 0.8% | 0.4% | 1.9% |
| 🔺 | L3 | 3.6 | 38.0% | 2.8% | 1.0% | 1.1% | 0.8% | 2.9% |
| 🟠 | N | 2.6 | 42.4% | 4.6% | 1.9% | 1.5% | 1.5% | 4.9% |
| 🔺 | L1 | 3.6 | 46.3% | 3.4% | 1.6% | 2.2% | 1.7% | 5.4% |
| 🔶 | U2 | 2.5 | 53.7% | 2.7% | 1.4% | 1.7% | 1.4% | 4.5% |
| 🔺 | L2 | 3.6 | 54.2% | 3.2% | 1.7% | 2.5% | 2.7% | 6.9% |
| 🔶 | U1 | 2.5 | 59.3% | 3.3% | 2.0% | 2.5% | 3.8% | 8.3% |

Public domain. CC0 1.0.
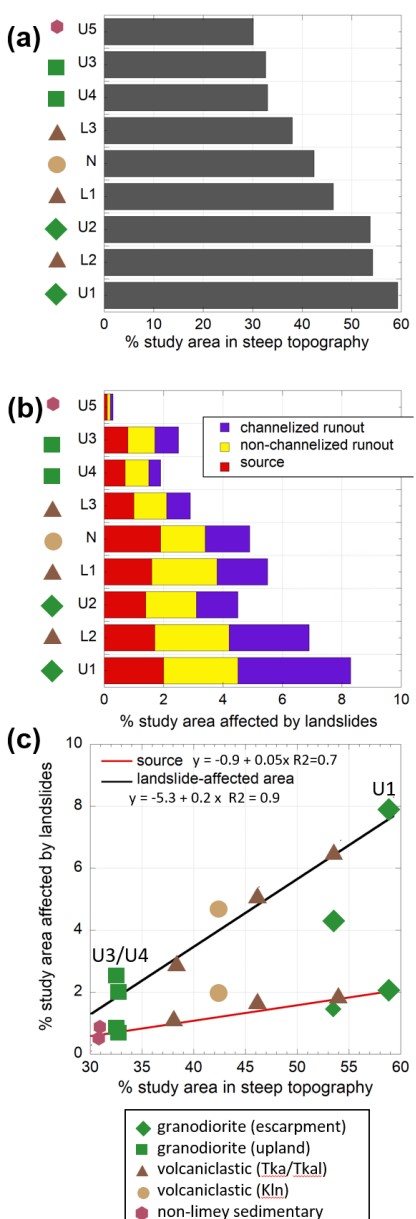

**Figure 8. a) Percentage of the nine individual study areas with steep topography. b) Percentage of individual study areas affected by Hurricane Maria landslides, divided into three mobility zones (Table 3). c) Linear regressions of percentage of study area susceptible to shallow landslides (column 2) with percentage of study area affected by 1) source zones (red line; column 4) and 2) total landslide-affected areas (source, non-channelized and channelized runout zones; column 7) (black line). Symbols shown represent geologic terranes and geomorphic terrains of study areas (Fig. 3 and Table 1).**

The percentage of each study area affected by landslide source zones (Table 3, column 4) increases slightly with the percentage of the study area susceptible to landslides (column 2). In contrast, the percentage of susceptible area affected by

Public domain. CC0 1.0.

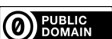



landslide sources (column 3) does not consistently increase or decrease. For example, study area N has the median percentage of area susceptible to landslides, but the highest (4.6%) percentage of area affected by landslide sources.

To examine trends in the area affected by landslides for the nine study areas, we fit a linear regression between the percentage of area susceptible to landslides with 1) source zones only and 2) total landslide-affected area. The linear regression produces slopes of 0.05 and 0.2, respectively (Fig. 8c). If all other contributing factors (e.g., rainfall distribution, material properties and hydrologic conditions) were equal, the percentage of susceptible area affected by landslide sources (Table 3, column 3) would be similar across all study areas, whereas the percentage of entire area affected by landslide sources (Table 3, column 4) would increase proportionally to area susceptible to landslides. In addition, if the area affected by landslide runout was directly proportional to the area affected by landslide sources, the slope of the regression lines would be equal. However, in comparison to the relation with source zones only, the percentage of total landslide-affected area increases at a greater rate than the percentage of area with steep slopes. In addition, the ratio between landslide-affected area (Fig. 8c, black line) and landslide source areas only (Fig. 8c, red line) shows the largest difference for study areas with the same underlying geologic terrane (Fig. 8c, U1 and U3/U4), where strength and hydrologic properties would likely be similar.

**5.3 Statistical distribution of topographic slopes within mobility zones**

We examined the statistical distributions and extracted percentile statistics ($P_{10}$, $P_{25}$, $P_{50}$, $P_{75}$, and $P_{90}$) (Fig. 9) for slopes in landslide source zones, non-channelized runout (predominantly runout on hillslopes), and channelized runout zones (Table 4). The median ($P_{50}$) represents typical slopes of Hurricane Maria landslide-affected areas, whereas $P_{10}$ represents characteristics of higher mobility landslides, with the ability to travel further downstream to areas of more gently sloping topography. Overall, these statistical distributions of slopes for the three zones show a progression of decreasing slopes along the travel path of landslides from Hurricane Maria.

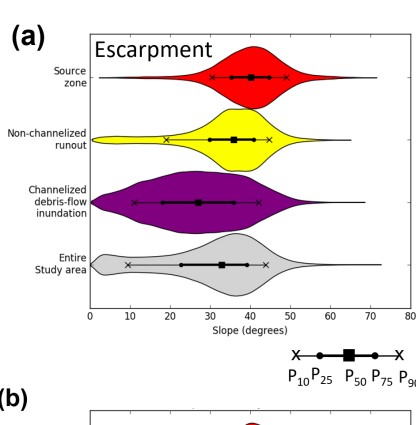

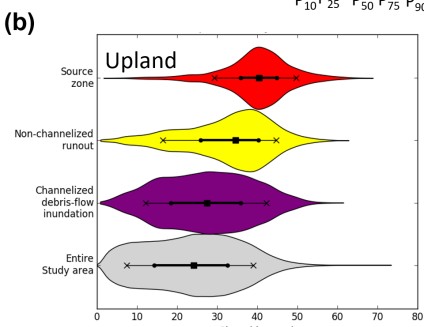

Public domain. CC0 1.0.


**Figure 9. Violin plots showing statistical distribution of slopes in the three zones of landslide mobility and in the entire study area for a) study area U1, a steep escarpment region and b) study area U3, a dissected, upland terrain. Both U1 and U3 are in a granitoid geologic terrane in Utuado. Symbols plotted within the violin plots show the interquartile range ($P_{25}$ to $P_{75}$), $P_{10}$ and $P_{90}$**

**Table 4. Percentile statistics of extreme ($P_{10}$) and average ($P_{50}$) slopes, in degrees, and the adjusted Fisher-Pearson coefficient of skewness ($G_1$) (Zwillinger and Kokoska, 2000) for slopes in the entire study areas and mapped landslide-affected areas, divided into the three zones of landslide mobility. The nine study areas are ordered by increasing percentage of area susceptible to landslides. This order also corresponds with the same order as the median slope of the entire study area (column 2).**

| symbol | study area name | entire study area | | | source zone | | | non-channelized runout | | | channelized runout | | |
|---|---|---|---|---|---|---|---|---|---|---|---|---|---|
| | | $P_{10}$ (1) | $P_{50}$ (2) | $G_1$ (3) | $P_{10}$ (4) | $P_{50}$ (5) | $G_1$ (6) | $P_{10}$ (7) | $P_{50}$ (8) | $G_1$ (9) | $P_{10}$ (10) | $P_{50}$ (11) | $G_1$ (12) |
| ● | U5 | 4.1 | 20.7 | -0.89 | 23.1 | 38.6 | 0.16 | 13.4 | 29.6 | -0.59 | 10.4 | 23.3 | -0.54 |
| ■ | U3 | 7.5 | 24.1 | -0.77 | 29.3 | 40.4 | 1.55 | 16.5 | 34.5 | -0.18 | 12.2 | 27.5 | -0.73 |
| ■ | U4 | 7.7 | 24.7 | -0.61 | 30.2 | 40.8 | 2.12 | 14.9 | 31.4 | -0.24 | 11.0 | 27.5 | -0.72 |
| ▲ | L3 | 12.1 | 26.7 | -0.41 | 25.5 | 36.7 | 0.72 | 16.0 | 29.9 | -0.37 | 13.6 | 27.8 | -0.51 |
| ● | N | 9.7 | 27.8 | -0.46 | 25.5 | 35.6 | 0.86 | 12.6 | 29.6 | -0.14 | 10.1 | 26.2 | -0.71 |
| ▲ | L1 | 10.1 | 28.6 | -0.70 | 29.0 | 40.0 | 1.75 | 21.9 | 37.5 | 0.52 | 17.0 | 34.1 | -0.46 |
| ◆ | U2 | 12.5 | 30.9 | -0.18 | 28.3 | 37.8 | 1.39 | 17.5 | 32.4 | 0.62 | 13.5 | 29.7 | -0.45 |
| ▲ | L2 | 11.8 | 31.4 | -0.43 | 31.6 | 41.1 | 2.56 | 23.8 | 39.0 | 1.08 | 17.9 | 34.7 | -0.47 |
| ◆ | U1 | 9.5 | 32.8 | -0.35 | 30.5 | 40.2 | 1.69 | 19.0 | 35.8 | 1.13 | 11.0 | 27.0 | -0.69 |

**Table 5. Minimum, maximum, mean, and standard deviation of $P_{10}$ and $P_{50}$ in the nine study areas (Table 4) for slopes (in degrees) in the entire study areas and for mapped landslide-affected areas, divided into the three zones of mobility.**

| | entire study area | | source zones | | non-channelized runout | | channelized runout | |
|---|---|---|---|---|---|---|---|---|
| | $P_{10}$ (1) | $P_{50}$ (2) | $P_{10}$ (3) | $P_{50}$ (4) | $P_{10}$ (5) | $P_{50}$ (6) | $P_{10}$ (7) | $P_{50}$ (8) |
| minimum | 4.1 | 20.7 | 23.1 | 35.6 | 12.6 | 29.6 | 10.1 | 23.3 |
| maximum | 12.5 | 32.8 | 31.6 | 41.1 | 23.8 | 39.0 | 17.9 | 34.7 |
| mean | 9.7 | 27.8 | 29.0 | 40.0 | 16.5 | 32.4 | 12.2 | 27.5 |
| standard deviation (σ) | 2.7 | 3.9 | 2.8 | 1.9 | 3.7 | 3.6 | 2.8 | 3.7 |

Variability between median ($P_{50}$) source-zone slopes in nine study areas was minimal, ranging from 35.6 to 41.1°, with a standard deviation (σ) of 1.9° (Table 5, column 4). In contrast, median slopes within the study areas, non-channelized and channelized runout zones (Table 4) showed almost twice the variability (σ = 3.9, 3.6, and 3.7°, respectively) (Table 5, columns 2, 6, 8). Extreme ($P_{10}$) values for study areas, source zones, non-channelized runout, and channelized runout (Table 4) displayed slightly higher variability (σ = 2.7, 2.8, 3.7, and 2.8°, respectively) between study areas (Table 5, columns 1, 3, 5, 7), compared

to $P_{50}$ of source-zones.

We quantified the asymmetry of the statistical distributions, using skewness ($G_1$) (Zwillinger and Kokoska, 2000). For overall study area slopes, skewness ranged from approximately symmetric (L3, N, U2, L2, U1) to moderately left skewed (U3, U4, U5, L1) (Table 4, column 3). The upland terrains (U3, U4) had left-skewed distributions of slopes. Slopes of source zones

Public domain. CC0 1.0.

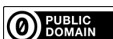
varied from approximately symmetric (U5) to moderately (L3, N) to highly right skewed (U1, U2, U3, U4, L1, L2) (Table 4, column 6). The distribution of slopes in areas of non-channelized runout was moderately left skewed to highly right skewed (Table 4, column 9) and channelized runout was approximately symmetric (L1, U2, L2) to moderately left skewed (U1, N, L3, U3, U4, U5) (Table 4, column 12). Our results indicate there was no clear pattern in slope characteristics of landslide-affected areas related to geologic terrane or geomorphic terrain.

Results also show median source-zone slopes were not correlated with median slopes of the study area (Fig. 10a); instead, they were relatively consistent across all study areas. There was some correlation between slopes of non-channelized runout zones (P$_{10}$ and P$_{50}$) and median slope of study area (Fig. 10b).

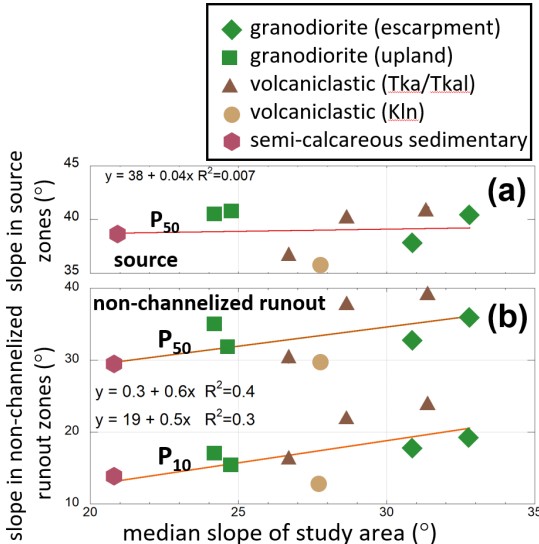

**Figure 10. Median slope of study area related to slopes of a) landslide source zones and b) non-channelized runout zones.**

### 5.4 Identification and characterization of channelized debris flows — Maria's most mobile (MMM)

We applied our two criteria to extract MMM from the published landslide inventories in the nine study areas. Figure 11 highlights landslides with > 40% of runout area located in close proximity (5 m) to a channel, including some non-channelized, lower mobility landslides (Fig. 10b). Figure 12 highlights MMM in the context of the two criteria.

Public domain. CC0 1.0.
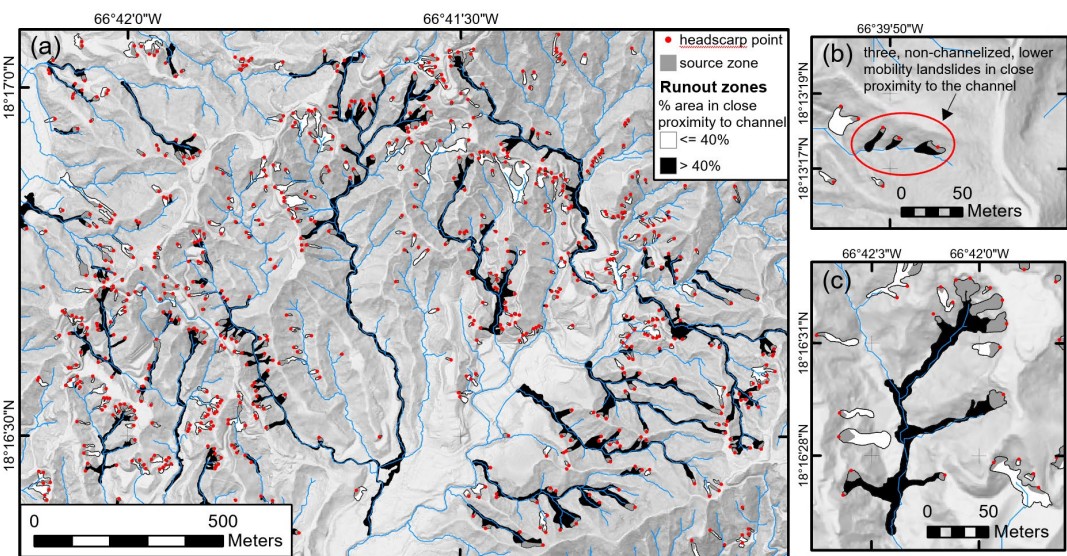

**Figure 11. Portrayal of some of Maria's most mobile debris flows. a) Map view of study area U1 showing 761 mapped landslide headscarp points (red points), associated source zones (dark gray), and 391 runout zones (Bessette-Kirton et al., 2019b; Einbund et al., 2021b). Runout zones with > 40% runout area located in or in close proximity to the channel are shown in black and those with < 40% in white. b) Zoomed-in view of study area U4, showing lower mobility landslides with short runout length, identified as having > 40% runout length in the channel. c) Zoomed-in view illustrates multiple landslide sources coalescing to a single debris-flow runout zone.**

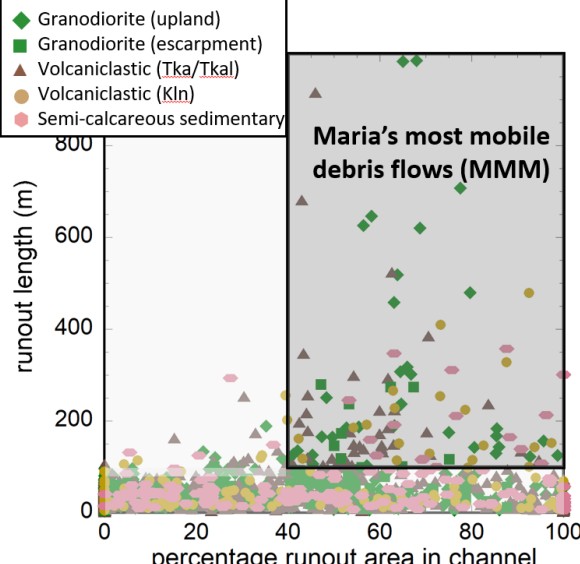

**Figure 12. Runout length and percentage of runout area in designated channels, for channelized runout zones in all study areas, grouped by geologic terrane. Gray box highlights MMM, identified by the characteristics of > 40% runout area in close proximity to the channel and > 100 m runout length.**

Table 6 summarizes the quantities and percentages of 1) landslide source areas (landslides) associated with MMM (columns 1,2, and 3) and 2) runout zones meeting the MMM criteria (columns 4, 5, and 6), where the influence of coalescence

Public domain. CC0 1.0.

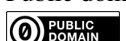



(Fig. 11c) results in multiple landslides contributing to a single runout path. Here, the runout zones may contain both non-channelized and channelized runout. We found that the percentage of landslide source areas associated with MMM ranged from 8.1% to 30% in different study areas. The study areas with three highest percentages, U1, N, and L2, are located in three different geologic terranes; here, 30%, 24.8% and 24.6% of landslides met the MMM criteria, respectively. The percentage of

runout zones meeting the criteria for MMM ranged from 3.3% to 9.0%, with the highest percentages found in three different geologic terranes, study areas U5, N, and U2; in this case the percentage of runout zones was 9.0%, 7.2%, and 6.7%, respectively. Study areas in other geologic terranes (L2, U1, U4, and L3) had slightly smaller percentage (5.2% to 5.6%) of runout zones identified as MMM; L1 and U3 had the smallest values, with 4.3% and 3.3% of runout zones identified as MMM. Our results highlight several observations for channelized debris flows triggered by Hurricane Maria: 1) channelized debris flows

are a minority of the landslides, 2) all geomorphic terrains and geologic terranes have some channelized debris flows, 3) escarpment terrains have a higher percentage of landslides associated with the MMM criteria, in comparison with upland terrains, 4) the percentage of runout zones does not show any consistent trends, and 5) there is no distinct pattern related to geologic terrane. Areas with a high landslide density, such as some of the escarpment terrains, may have many landslides in close proximity, thereby increasing the potential to coalesce in the nearest drainage. Given the influence of coalescence, the

percentage of landslides and percentage of runout zones (Table 6, column 3 and 6) did not correlate.

**Table 6. Number and percentage of landslides and runout zones meeting the most mobile (MMM) criteria for nine study areas, sorted by percentage of landslides associated with MMM.**

| | | # landslides (1) | # landslides associated with MMM (2) | % landslides associated with MMM (3) | # runout zones (4) | # runout zones with MMM criteria (5) | % runout zones with MMM criteria (6) |
|---|---|---|---|---|---|---|---|
| ◆ | U1 | 761 | 228 | 30.0% | 391 | 21 | 5.4% |
| ● | N | 440 | 109 | 24.8% | 263 | 19 | 7.2% |
| ▲ | L2 | 480 | 118 | 24.6% | 306 | 17 | 5.6% |
| ▲ | L3 | 288 | 57 | 19.8% | 210 | 11 | 5.2% |
| ◆ | U2 | 382 | 71 | 18.6% | 238 | 16 | 6.7% |
| ▲ | L1 | 525 | 87 | 16.6% | 375 | 16 | 4.3% |
| ■ | U3 | 191 | 29 | 15.2% | 90 | 3 | 3.3% |
| ⬡ | U5 | 168 | 20 | 11.9% | 156 | 14 | 9.0% |
| ■ | U4 | 124 | 10 | 8.1% | 130 | 7 | 5.4% |
| | all | 3235 | 719 | 22.2% | 2159 | 124 | 5.7% |

Our results from 124 debris-flow inundation zones (Table 7) show that MMM typically ($P_{50}$) occur in stream reaches

with stream order ≤ 2; all MMM occur in stream reaches with stream order ≤ 5 (Table 7). MMM have mean stream slopes of about 5 to 16°, mean planform curvature of 0.03 m$^{-1}$, and a mean $P_{src}$ of 58% to 85%. Less common, but more extreme

Public domain. CC0 1.0.


endmembers of MMM have a stream order of 4 or 5, stream slope of 0.2 to 7.0°, planform curvature from a smoothed DEM of

0.0 to 0.02 m$^{-1}$, and $P_{src}$ of 9% to 33%. These results provide criteria to define debris-flow growth zones for inundation scenarios.


**Table 7. Percentiles of maximum Strahler stream order, mean stream slope, mean planform curvature, and mean percentage contributing area susceptible to shallow landslides ($P_{src}$) for debris-flow inundation zones. Values are based on the mean or maximum (for stream order) value along runout path of MMM.**

| parameter | percentile | Lares (L1, L2, L3) | Utuado (U1, U2, U3, U4) | Naranjito (N) | Utuado (U5) |
|---|---|---|---|---|---|
| maximum Strahler stream order | $P_{50}$ | 2 | 3 | 2 | 2 |
| | $P_{75}$ | 3 | 3 | 3 | 3 |
| | $P_{90}$ | 4 | 4 | 4 | 4 |
| | $P_{100}$ | 5 | 5 | 4 | 4 |
| mean stream slope (degrees) | $P_{50}$ | 15.9 | 9.8 | 15.0 | 4.6 |
| | $P_{25}$ | 11.2 | 7.0 | 11.2 | 1.7 |
| | $P_{10}$ | 8.2 | 4.8 | 8.8 | 0.9 |
| | $P_0$ | 3.1 | 1.1 | 7.0 | 0.2 |
| mean planform curvature (m$^{-1}$) | $P_{50}$ | 0.03 | 0.03 | 0.03 | 0.03 |
| | $P_{25}$ | 0.02 | 0.02 | 0.02 | 0.01 |
| | $P_{10}$ | 0.02 | 0.01 | 0.01 | 0.01 |
| | $P_0$ | 0.01 | 0.01 | 0.01 | 0.00 |
| mean percentage contributing area susceptible to shallow landslides ($P_{src}$) | $P_{50}$ | 85% | 79% | 58% | 75% |
| | $P_{75}$ | 77% | 65% | 42% | 52% |
| | $P_{90}$ | 62% | 55% | 20% | 29% |
| | $P_{100}$ | 33% | 29% | 9% | 14% |

Public domain. CC0 1.0.



## 6 Results of runout and inundation modeling

### 6.1 *H/L* scenarios

For the mapped study areas, consisting predominantly of highly dissected topography, steep slopes, and narrow valleys, even large changes in α did not greatly modify zones of *H/L* runout. Figure 13 illustrates where there is some minimal additional runout area with a substantial decrease in α from 30 to 20°. Significant areas on the hillslope are encompassed by *H/L* runout zones in the range of 25 to 30° (Fig. 13, brown zones) derived from Hurricane Maria source areas, whereas, when α is < 20°, the affected areas (Fig. 13, blue zones) are located within narrow channel bottoms. A decrease to 20° captures additional area within the non-channelized runout zones of mapped landslides, without a significant increase in areas identified as susceptible (added yellow areas). The area shown in figure 13 is representative of the majority of areas where landslide inventories were available. Appropriate choice of α for regional maps is controlled by the slope angle of topography upslope of channels, quantified in our analysis of slopes in mapped non-channelized runout zones (Table 4, columns 7 and 8). Our analysis of slopes in these zones indicates a wide range of potential α values. To eliminate the potential for gaps between estimated non-channelized runout areas and channelized debris-flow inundation areas, we selected α = 20° for our susceptibility maps.

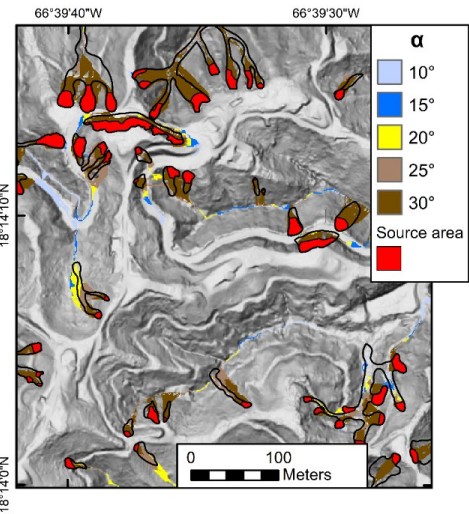

**Figure 13. *H/L* runout results for a range of α values (10, 15, 20, 25, and 30°) in a section of U2, an escarpment study area in the granitoid terrane, Utuado. Source areas from Einbund et al., 2021b.**

### 6.2 Debris-flow inundation scenarios

Our evaluation of eight debris-flow inundation scenarios (Fig. 6) indicates that modification of the parameters defining debris-flow growth zones can have a significant influence on the pattern and extent of inundation. For example, purple zones in Fig. 14a illustrate the inundation area where debris-flow growth zones extended lower in the drainage network, as controlled by a larger value for maximum stream order and smaller value for minimum stream slope. Maximum volume ($V_{max}$ = 1000 m³) and growth factor ($c_1$ = 0.01 m³ m⁻²) are held constant in scenarios A-1k, B-1k, C-1k, and E-1k. In the case of relatively wide basins and multiple incoming tributaries, more generous growth zones

Public domain. CC0 1.0.
create greater runout lengths (Fig. 14a). In the same topography, increased growth factors with a fixed growth zone also create longer and wider inundation zones (Fig. 14b).

In contrast, narrower drainage basins with few contributing tributaries that abruptly exit steep mountainous terrain into a wide, flat valley over a short distance, exhibited no difference between results with highly variable definition of growth zones. Fig. 14c demonstrates this situation, where scenario B-1k, C-1k, and E-1k produce identical inundation results. Scenarios C-1k, C-3k, and C-10k, with increasing growth factor ($c_1$ of 0.01, 0.05 and 0.2 $m^3\ m^{-2}$, respectively) and $V_{max}$ (1000, 3000, and 10,000 $m^3$, respectively) produce progressively wider and longer runout length regardless of basin shape (Figs. 14c, d), where increased $c_1$ produces wider inundation zones higher in the drainage network. Increased $V_{max}$ can produce both wider and longer inundation zones. In these scenarios, debris-flow growth will always be halted
when the maximum stream order or minimum stream slope criteria for a given scenario is achieved, yielding a volume proportional to the upslope area susceptible to landsliding. Therefore, growth zones that terminate before $V_{max}$ is achieved will have smaller areas of inundation.

**Figure 14. Results from debris-flow inundation scenarios with varied growth controls in different topographies. a) Wider topography with varied growth zones. A-1k, B-1k, C-1k, and E-1k showing progressively greater inundation lengths with more generous growth zones. b) Wider topography with increasing growth factors and maximum volumes. C-1k, C-3k, and C-10k showing progressively**

Public domain. CC0 1.0.
wider and longer inundation lengths with increased growth factor and maximum volume ($V_{max}$). c) Narrower basin with fewer incoming tributaries and varied growth zones. Here, B-1k, C-1k, and E-1k produce identical results, only A-1k differs. d) Narrower basin with increasing growth factors. Regardless of basin shape, C-1k, C-3k, and C-10k produce progressively wider and longer

inundation lengths. Black outlines are mapped landslides from Hurricane Maria including source areas (Einbund et al., 2021b). The scale bar is the same for all panels.

**6.3 Evaluation of predictive success for debris-flow inundation scenarios**

Figure 15 shows our ROC analysis for the eight scenarios, evaluated for all MMM. Solid gray lines show positive likelihood ratio ($PLR = TPR/FPR$), where higher $PLR$ indicates a higher likelihood of correct prediction. Dashed gray lines show distance

from the upper left corner, the location of perfect classification. False positive rate ($FPR$) may be over-estimated in cases where the actual terminus of debris-flow deposits could not be identified.

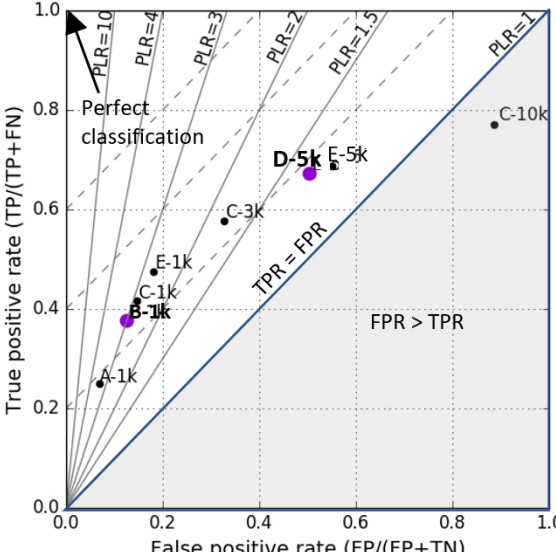

Figure 15. Receiver Operator Characteristic (ROC) plot for eight debris-flow inundation scenarios evaluated for nine study areas affected by MMM. The two scenarios selected for regional susceptibility maps are highlighted in purple. Scenario B-1k is a likely

scenario and D-5k is a less likely, but more hazardous scenario.

For our regional susceptibility maps, we first selected a scenario (B-1k) with relatively high value for $PLR$ (Fig. 15, $PLR$ = ~3). This scenario, B-1k, defines zones of extremely high susceptibility to debris-flow inundation. Scenario B-1k minimizes over-prediction, as characterized by relatively low $FPR$ and high $PLR$. $TPR$ based on the area affected is ~0.38, whereas consideration of the number of debris-flow inundation runout zones provides more impressive $TPR$ values, ranging

between 0.50 and 0.95 (Table 8). $TPR$ was lowest for U5, the largest study area with a very low percentage of area affected by landslides (Table 3). Overall, for all study areas combined, scenario B-1k, identified 85% of debris-flow inundation zones.

To aid selection of a more extensive scenario, we examined scenarios in map view, in combination with a ROC plot (Fig. 15). We selected D-5k because it provides an increased true positive rate ($TPR$) before the significant decrease in positive likelihood ratio ($PLR$) seen with scenario E-5k. Scenario D-5k identified 90% ($TPR$ = 0.90) of debris-flow runout zones (Table

8). This $TPR$ of 0.90 is in alignment with the $TPR$ selected for source-area susceptibility thresholds (Baum et al., 2024). Scenario D-5k defines a more hazardous, but less likely, scenario representative of the area affected in the most severely impacted drainages during Hurricane Maria.

Public domain. CC0 1.0.

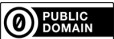



**Table 8. Number of debris-flow inundation zones (MMM) and *TPR* for the two selected susceptibility scenarios. MMM are grouped by geologic terrane.**

| | # debris-flow inundation zones (MMM) | *TPR* for number of detected debris-flow inundation zones | |
|---|---|---|---|
| | | **B-1k** | **D-5k** |
| Lares (L1, L2, L3) | 44 | 0.95 | 0.95 |
| Utuado (U1, U2, U3, U4) | 47 | 0.83 | 0.89 |
| Utuado (U5) | 14 | 0.50 | 0.71 |
| Naranjito (N) | 19 | 0.89 | 0.89 |
| All study areas combined | 124 | 0.85 | 0.90 |


### 6.4 Susceptibility maps portraying three mobility zones

We applied our linked-model approach to create regional susceptibility maps delineating potential locations of landslide initiation, downslope runout, and debris-flow inundation during prolonged, intense rainfall, for Lares, Naranjito, and Utuado municipalities, encompassing a total area of 560 km$^2$. Potential source areas (initiation) from shallow landslide susceptibility

modeling (Baum et al., 2024) were used to identify 20° *H/L* runout zones and upslope contributing source areas for volume estimations (Eq. (4)) used in debris-flow inundation scenarios B-1k and D-5k.

In our regional susceptibility maps (e.g., Fig. 16), debris-flow inundation areas (purple zones) overlie all other zones and may conceal underlying source (red) and *H/L* runout zones (yellow); non-channelized runout zones (*H/L*) underlie all other colors in the perspective view. Debris-flow inundation zones are shown in two shades of purple, where dark purple is scenario B-

1k, highlighting inundation in upper parts of the drainage network, and light purple is scenario D-5k. In steep, dissected escarpment terrains, such as U1, most of the topography meets the criteria for source zones, resulting in substantial overlap between areas susceptible to shallow landslides and *H/L* runout zones.

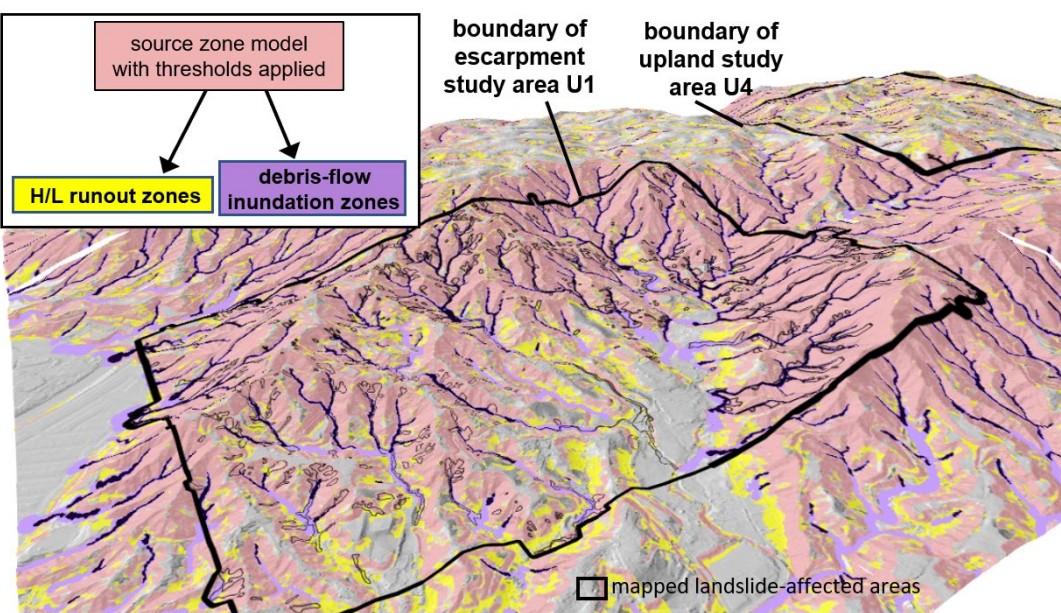

Public domain. CC0 1.0.

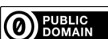
**Figure 16. Perspective view of landslide susceptibility results in part of Utuado encompassing study area U1 (2.5 km², located on an
escarpment, and study area U4, located in an upland terrain. Mapped landslides (Bessette-Kirton et al., 2019b; Einbund et al., 2021b)
shown for reference with black outlines. Dark purple is scenario B-1k and light purple is the more extensive scenario D-5k.
Approximate location at center of image is 18° 16′ 40″ N, 66° 41′ 10″ W.**

In open-slope topographies, the absence of channelization controls the applied modeling approach and resulting
susceptibility. Figure 17 shows a comparison of landslide-affected areas (Fig. 17a) and modeling results (Fig. 17b). Here,
susceptibility results show $H/L$ runout for non-channelized areas; debris-flow inundation is not modeled.

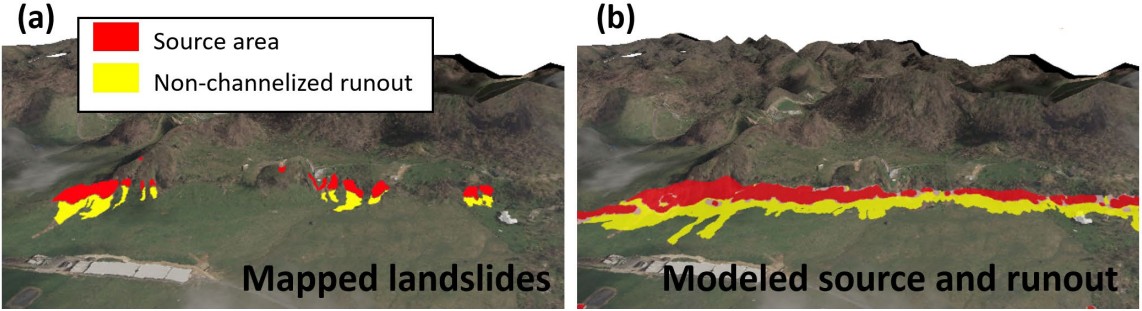

**Figure 17. Perspective view of post-Maria aerial imagery draped on DEM in a non-channelized open-slope topography in Northern
Utuado. a) Mapped landslides located in semi-calcareous sedimentary units adjacent to karst topography in northern Utuado
(Baxstrom et al., 2021a) divided into source and non-channelized runout, b) Runout modeling results of areas susceptible to shallow
landslides (Baum et al., 2024) and $H/L$ runout zones (yellow). Approximate location at center of image is 18° 18′ 5″ N, 66° 49′ 0″ W.**

## 7 Discussion

We used three zones of landslide mobility as the framework for topographic analysis and a linked-model approach to estimate
areas susceptible to landslide runout. Both non-channelized and channelized landslide runout were observed in Hurricane Maria
(Figs. 1, 4, 7) and have the potential to adversely impact roads and infrastructure.

### 7.1 Insights from topographic analysis

Our topographic analysis of landslide-affected areas quantified the area affected within each mobility zone and provided
statistics describing the topographic slopes in each zone. Study areas with the highest percentage of area susceptible to landslides
had the largest area affected by landslides during Maria (Table 3, Fig. 8). The percentage of total landslide-affected area to
percentage of study area with steep slopes (Fig. 8b,c) increases at a greater rate (from 0.3% to 8%; Table 3, column 7) than the
more modest increase (from 0.1% to 2%; Table 3, column 4) of the percentage of study area affected by landslide source areas
only. This disproportionate increase in total affected area with a high percentage of steep slopes suggests increased mobility and
greater hazard in these areas, in contrast to more isolated steep slopes. The statistical distribution of topographic slopes shows
that the escarpment study areas (U1 and U2) did not have steeper source areas than upland terrains (Table 4, column 4 and 5),
although the overall slopes are steeper in the escarpment areas. These combined observations indicate differences are not due
solely to slope angle, material strengths, hydrologic properties or conditions, but also to predisposing factors related to
geomorphic setting. Possible explanations for this phenomenon might include coalescence (e.g., Coe et al., 2021), more readily
available entrainable material due to frequent landslides (e.g., Coe et al., 2021; Scheip and Wegmann, 2022), soil depths, and/or
topographic controls (e.g., Corominas, 1996; Coe et al., 2011). Escarpment terrains, with more terrain susceptible to landslides,

Public domain. CC0 1.0.
provide more opportunity for the coalescence of contributing landslide source areas over a greater length of runout path, either channelized or non-channelized. Debris-flow runout paths may traverse a substantial distance through areas of additional contributing landslide source area. In addition, individual basins in escarpment terrains (Fig. 18a) have higher topographic relief than dissected uplands (Fig. 18b). Higher relief provides the potential for greater runout lengths before there is a change in stream slope conducive to deposition — this results in correspondingly larger runout areas (Fig. 8). Likewise, areas with high

drainage density will provide greater opportunity to amalgamate multiple flows, potentially having a nonlinear impact on access to readily entrainable material.

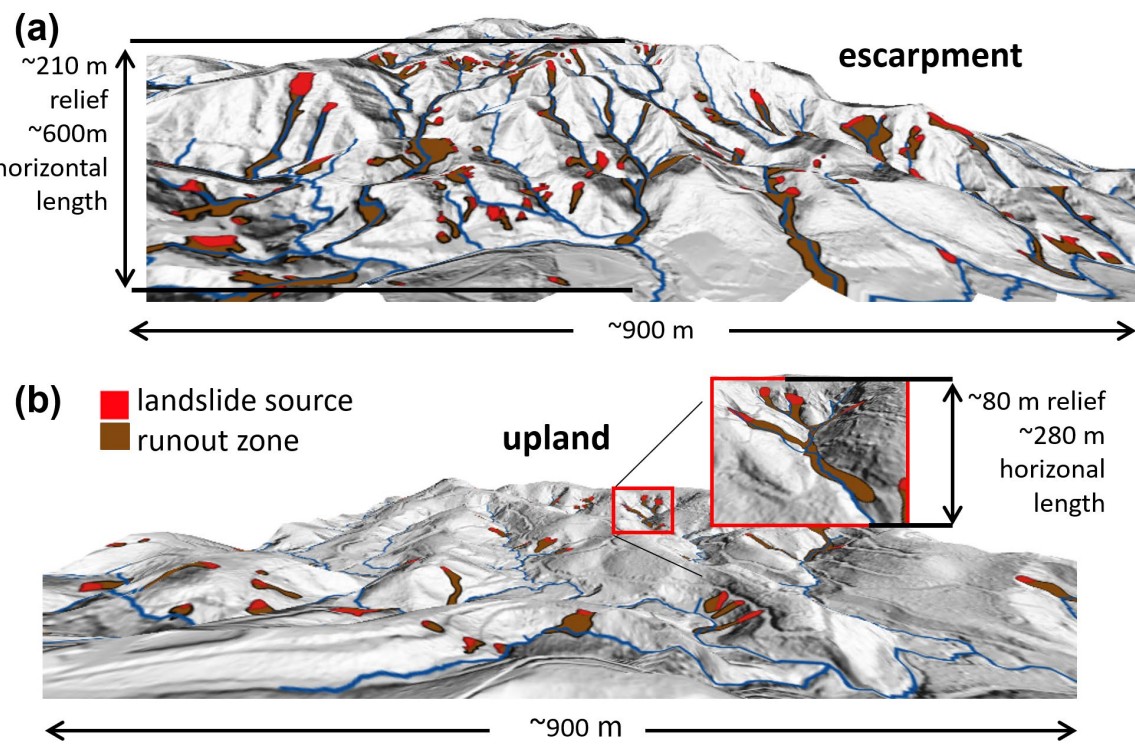

**Figure 18. Perspective views showing mapped landslide source and runout zones caused by Hurricane Maria in Utuado, Puerto Rico**

**(data from Einbund et al., 2021b): a) escarpment terrain in study area U1, and b) upland terrain in study area U3. Approximate location of a at center of image is 18° 16′ 30″ N, 66° 41′ 20″ W; b is located at 18° 16′ 30″ N, 66° 47′ 35″ W.**

### 7.2 Mobility metrics

    Mobility metrics can be used to quantify relative landslide mobility or predict the potential for future mobility. At initiation, the

potential for landslide mobility is controlled by material properties, pore pressures, and initial porosity of the material (e.g., Iverson, 1997; Iverson et al., 2000; Reid et al., 2008; Iverson et al., 2011; Collins and Reid, 2020). Mobility can be further enhanced by relatively high initial moisture content of the material over which a debris flow travels (Iverson et al., 2011; Reid et al., 2011), proximity to a channel network (e.g., Coe et al., 2011), and channel confinement (e.g., Iverson, 1997). In addition, topographic constraints such as height of fall, regularity of the pathway, bends, deflections, and confinement can limit or enhance

landslide travel distance (e.g., Corominas, 1996).

Public domain. CC0 1.0.

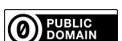



Several mobility metrics have been discussed in the literature, including: unitless runout number ($L/A^{1/2}$) (Wallace et al., 2022), $L$, excessive travel distance ($L_e$) (Hsu, 1975; Corominas, 1996), relative excessive travel distance ($L_r$) (Corominas, 1996), $H/L$, angle of reach (α), $L/H$, and $V/A^{2/3}$ (e.g., Scheidegger, 1973; Hsu, 1975; Nicoletti and Sorriso-Valvo, 1991; Corominas, 1996; Iverson et al., 2015; Legros, 2002; Wallace et al., 2022). In Puerto Rico, the percentage of landslide-affected area (Table 3, column 7 and Fig. 8) serves as a metric to compare overall landslide mobility between study areas.


The two debris flows shown in Fig.18 illustrate one potential problem with the use of $H/L$ for assessing mobility. For the escarpment flow (Fig. 18a), $L = \sim600$ m and α = 19°, whereas for an upland terrain debris flow (Fig. 18b), $L = 280$ m and α = 16°. Lower α suggests greater relative mobility for dissected upland flow, despite its significantly shorter runout length. Thus, it can be difficult to determine if there is a fundamental difference in the initial potential for mobility (as measured by any metric) or the ultimate mobility; rather, each debris flow traveled until a decrease in stream slope, sufficient for deposition, was reached.


In these Hurricane Maria examples, topographic relief defines the $H$ and $L$ of travel before a decrease in stream slope thereby controlling the ultimate mobility of these flows.

In our linked-model approach, topographic factors are automatically incorporated via multiple mechanisms. $H/L$ runout zones are inherently controlled by local topography and in channelized topography, minimal additional runout area is modeled


with decreased α (as illustrated in Fig. 13). In channelized topography, debris-flow growth is restricted to drainages with sufficient stream slope and greater than 20% of upslope contributing area susceptible to shallow landslides ($P_{src}$). These restrictions, thereby provide a self-regulating method to estimate potential inundation. Likewise, most channelized debris flows will continue along their pathway until reaching a significant decrease in channel slope, after which deposition is predominant. This concept is applied in our inundation modeling, without the need to define spatially variable input parameters.

**7.3 Considerations for linked-model approach**

**7.3.1 Selection of angle of reach**

Although much debate exists in the literature regarding the application of the angle of reach (α) (e.g., Hungr et al., 2005), $H/L$ is commonly used to quantify relative mobility of landslides. As noted by Wallace and Santi (2021), there are potential limitations to the usefulness of $H/L$ to describe landslide mobility. Specifically, unless there is a change in gradient downslope of the source,


$H/L$ only measures the overall slope gradient and does not distinguish between short- and long-runout events on uniform slopes. We found a correlation between slopes of non-channelized runout zones ($P_{10}$ and $P_{50}$) and median ($P_{50}$) slope of study area (Fig. 10b), an indication that local topography can influence H/L runout angles, an important factor to consider when evaluating statistics of runout angles from landslide inventories. Herein, we do not use $H/L$ or $L$ to compare mobility between landslides as these metrics are also highly dependent on basin shape, relief of basin, angle of intersection with tributary junctions, and type of


landslide (e.g., Corominas, 1996). However, the subtleties of $H/L$ measurements complicate the selection of α for our modeling application. To address these complications, we assessed a wide range of $H/L$ values at 5° increments of α (Fig. 13) and considered the amount and locations of additional runout area. In many of the study areas with steep, dissected topography, we observed that most non-channelized runout is encompassed by $H/L$ runout zones identified with α > 25°. Some additional non-channelized runout is within the zone between 20° and 25°. Additional areas encompassed by a value less than 20° typically are


located in the channel, where our methods for debris-flow inundation were applied. In open-slope topographies, where very few mapped landslide-affected areas were available for quantitative assessment, our choice of α = 20° compares well visually with observed runout from Hurricane Maria (Fig. 17).

In addition, world-wide datasets of $H/L$ (e.g., Corominas, 1996) provide useful relative comparison of some the most notable documented landslides in the literature. Many types of landslides, including rockfalls, translational slides, debris flows,

Public domain. CC0 1.0.

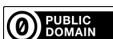



earthflows, mudslides, and rock avalanches, display a reduction in α with increasing volume (e.g., Corominas, 1996; Iverson et
al., 2015). The further application of reduced α with increasing volume can be employed with our methods to refine estimates of
areas susceptible to non-channelized runout.

### 7.3.2 Selection of debris-flow growth zones

Selection of debris-flow growth zones controls where debris-flow growth factors are applied. The location of these zones can

impact inundation patterns and extent as significantly as the choice of $c_1$ or $V_{max}$ (Fig. 14). In some cases, stream slope can serve
as a single control on location of these growth zones. Reid et al. (2016) summarized slopes where deposition was predominant
from previous studies, with deposition for confined flows occurring at slopes less than 15°, and in some cases as low as 1°. More
recent studies provide stream-slope thresholds in a similar range. For debris flows in North Carolina, Scheip and Wegman (2022)
found a transition from erosion to deposition in the range of 8° to 30°, with a mean value of 18°. Burns et al. (2022) identified

that debris-flow transport (non-deposition) occurs either on steep (> 8°) channel reaches or highly confined reaches with gentler
slopes (< 5°). In Puerto Rico, field measurements of locations where growth transitioned to deposition (3° to 8°) guided our
selection of stream slopes for growth zones (Coe et al., 2021).

In the diverse topography of Puerto Rico, complexities such as short segments of steep stream slopes distanced from
areas of landslide susceptibility, and locally isolated areas susceptible to landslides, necessitated multiple parameters to restrict

growth zones. We used characteristics of MMM inundation zones to constrain growth zones for our debris-flow inundation
modeling. Statistics based on analysis of MMM provided ranges of values for Strahler stream order, percentage of contributing
area susceptible to debris flow ($P_{src}$, Eq. (1)), and planform curvature (Table 7). Stream reaches with high values of $P_{src}$, are
directly coincident with locations where the most susceptible areas are located, and this parameter might be used as the primary
control to define growth zones.

### 6.3.3 Selection of debris-flow growth factors and volumes

Volume estimates within zones of debris-flow growth are controlled by debris-flow growth factors and limited by $V_{max}$ .
Published values for growth factors (sometimes termed "growth rates") are typically estimated from differences in elevation
calculated from photogrammetry or lidar-derived DEMs (Reid et al., 2016; Coe et al., 2021; Scheip and Wegman, 2022). Reid et
al. (2016) summarizes published length-based growth factors and applies length- and area-based growth factors for Oregon, with

values ranging from 11–24 $m^3$ $m^{-1}$ and 0.12–0.2 $m^3$ $m^{-2}$, respectively.

For Hurricane Maria, estimates of $c_1$ and $V_{max}$ based on difference of DEMs from pre- and post-Maria lidar were
available (Coe et al., 2021). Our calculation of growth factors normalized to contributing susceptible area rather than full
contributing area (Table 2) allows the application over large areas, where susceptibility to shallow landslides is spatially variable.

For our assessment of debris-flow inundation scenarios, we found that the scenario with the largest maximum volume

estimate (C-10k), resulted in a false positive rate that exceeded the true positive rate, indicating severe overprediction (Fig. 15).
The maximum volume for C-10k originated from a site with significant volume contribution from a single landslide and minimal
contribution from debris-flow growth mechanisms (Four Car site, Coe et al., 2021). In this situation, our area-integrated growth
factor will under-estimate volume at the site of channel initiation and over-estimate growth along the travel path. In regions
where these characteristics are known to be the predominant pattern, smaller growth factors, power-law growth factors, or initial

source volumes can be applied to our methodology.

Future investigations of debris-flow growth factors could help determine the applicability of growth-factors beyond the
specific basins for which they were calculated. Likewise, questions related to whether the same basins will repeatedly generate

Public domain. CC0 1.0.

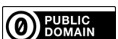



debris-flows of the same magnitude or may have a delay in time after a storm event has removed readily entrainable material from the channels is an important consideration. In addition, the susceptibility of different drainage basins with seemingly

similar characteristics may depend on human modifications within each basin.

### 7.3.4 Assessment of debris-flow inundation scenarios

Contingency table metrics provide multiple evaluation criteria (e.g., Powers, 2011). Choosing a metric for optimization of scenarios depends on the objectives — selection of a high value for *TPR* maximizes the number of true positives (*TP*) and provides high success in prediction of susceptible areas. Unfortunately, selection of a scenario based solely on *TPR* typically

contributes to a higher value for *FPR* (Fig. 15) and can result in assignment of significant area not affected by actual landslides in a given landslide-inducing event as susceptible. In the case of landslide modeling, although this may be a false prediction for a specific previous event, the results may produce successful prediction in a future event. Typically, only a small percentage of area susceptible to landslide initiation or runout is affected by a single event. For example, in Hurricane Maria, only 0.4% to 3.3% of steep slopes were subject to landslides (Table 3).

When available, information regarding the specific locations of landslide initiation and stream reaches where readily entrainable material is available can be incorporated into our methods. For example, Jibson (1989) noted channel scouring and side-slope debris contributed 90–95% of debris-flow volume in debris flows along the south-central coast of Puerto Rico, during a tropical storm on October 5–8, 1985. Although generalized parameters provide a good initial estimate, field observations and debris flow history can focus on locations where conditions conducive to enhanced debris-flow growth are present. Field

observations have the potential to highlight basins with elevated level of hazard and improve predictive success of modeling results.

### 7.4 Limitations of linked-model approach

Our methods are not a replacement for site-specific investigations or cases where physics-based models can be calibrated to provide more detailed information, including estimates of velocity and inundation depth (e.g., McDougall and Hungr, 2004;

Christen et al., 2010; George and Iverson, 2014; Iverson and George, 2014; FLO-2D Software Inc., 2017; Barnhart et al., 2021). Our methodology is designed to assess large regions for runout and debris-flow inundation hazard. In areas of high concern, field studies, analysis of past events, and application of physics-based models may provide more refined hazard estimates.

The advantages of our modeling approach include the ability to estimate areas susceptible to runout and inundation without the need to invoke spatially variable angles of reach, debris-flow growth zones, or debris-flow growth factors based on

material properties. Our linked-model approach successfully estimated runout susceptibility for three municipalities in Puerto Rico, where knowledge of site-specific materials and conditions was limited.

### 8 Conclusions

Our analysis of landslide-affected areas from Hurricane Maria illustrates that both non-channelized and channelized landslide runout (debris-flow inundation) occurred across nine study areas, encompassing escarpment and upland terrains in volcaniclastic,

granitoid, and non-limey sedimentary geologic terranes. Non-channelized runout was the most recurrent, whereas channelized runout was the most areally extensive. Using the concept of zones of mobility, we analyzed topographic characteristics of landslide-affected areas and applied an empirical, linked-model approach to estimate areas susceptible to non-channelized and

Public domain. CC0 1.0.

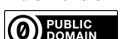



channelized runout. Our linked-model approach provided a self-regulating method, whereby topography controls the runout method, spatial distribution, and extent of potential landslide runout and debris-flow inundation zones in four primary ways:

1. The presence or absence of a channel network controls the application of runout method, where areas susceptible to non-channelized runout are identified by a minimum angle of reach and channelized debris-flow inundation zones are estimated using debris-flow growth factors combined with volume-area relations.

    2. $H/L$ runout zones provided a transition from source zones to channels and identified non-channelized runout in areas with open-slope topography, where channels are not present.

3. In channelized topography, debris-flow growth zones were restricted to steep stream reaches (> 5°) possessing characteristics of Hurricane Maria's most mobile debris flows (MMM): low stream order (≤ 4), high percentage of contributing area susceptible to debris flows (> 20% $P_{src}$), and concave planform curvature (< 0.02 m$^{-1}$).

    4. Within the zones of debris-flow growth, volumes were calculated as a function of upslope area susceptible to shallow landslides, whereby drainage basins with minimal susceptible area are assigned smaller volumes and highly susceptible

areas are assigned larger volumes, up to a specified maximum, $V_{max}$. The rate of debris-flow growth is controlled by a growth factor, $c_1$.

Our coupled-model approach incorporates these methods for portraying runout and inundation for landslides over a range of mobility and enables runout assessment over large regions without the computational effort required by physics-based models or the need to identify precise locations and volumes of landslide sources. To provide an assessment of areas susceptible to

landslide runout and inundation, we applied our two runout models in three municipalities that had high landslide density from Hurricane Maria: Utuado, Lares, and Naranjito, covering a total area of 560 km$^2$.

Our results illustrate that that geomorphic setting can exert a primary influence on debris-flow runout. Escarpment terrains, with high relief and a high percentage of contributing area susceptible to shallow landslides, were predicted to have larger areas affected by long-runout debris flows in contrast to dissected-upland terrains. These patterns match observations from Hurricane

Maria. Assessment of the predictive success of our debris-flow inundation modeling, based on 124 debris flow runout zones from Hurricane Maria's most mobile debris flows in all terrains, demonstrates that one of our scenarios identified 90% of the Hurricane Maria debris-flow runout zones.

**Code availability**

Computer codes used for this study are available in a U.S. geological Survey software repository (Cronkite-Ratcliff et al., 2024).

**Data availability**

Pre- and post- Hurricane Maria lidar-derived DEMs are available through the national map at https://apps.nationalmap.gov/lidar-explorer/ (U.S. Geological Survey 2018, 2020a,b,c). Landslide inventories are available as U.S. Geological Survey Data Releases (Bessette-Kirton et al., 2019b; Baxstrom et al., 2021a, 2021b; Einbund et al., 2021a, 2021b). Hurricane Maria's Most Mobile (MMM) landslides are also available as a U.S. Geological Survey Data Release (Brien et al., 2024).

Public domain. CC0 1.0.

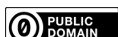



## Author contributions

DLB performed linked-model simulations with input from all authors. DLB designed and conducted topographic analysis. MER managed the Puerto Rico project subtask. CC contributed statistical expertise and wrote the computer code. JPP performed bandpass filtering of DEMs. DLB wrote and edited the manuscript. All authors reviewed the manuscript and provided comments.

## Competing interests

The contact author has declared that none of the authors has any competing interests.

## Disclaimer

Any use of trade, firm, or product names is for descriptive purposes only and does not imply endorsement by the U.S. Government. All authors commented on previous versions of the manuscript.

## Acknowledgements

This work was supported in part by the Additional Supplemental Appropriations for Disaster Relief Requirements Act, 2018 (P.L. 115–123). We thank Jeffrey Coe and Katy Barnhart for their time and thoughtful reviews. DLB thanks those who provided opportunities, encouragement, and/or positive feedback, including Phillip Dawson, Judy Fierstein, Dean Miller, and William Schulz.

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
