# Peer review of "Topographic controls on landslide mobility: Modeling hurricaneinduced landslide runout and debris-flow inundation in Puerto Rico"

_Natural Hazards and Earth System Sciences, 2024_

## Author Comment (AC3)

Response to Anonymous Referee #3 comments on 'Topographic controls on landslide mobility: Modeling hurricane-induced landslide runout and debris-flow inundation in Puerto Rico' by Dianne L. Brien, Mark E. Reid, Collin Cronkite-Ratcliff, and Jonathan P. Perkins

reviewer comments in grey and replies in black text.

This research is attempting to develop a model that combines the model for the landslide initiation and the sediment flow down as debris flow. In addition, two models are combined for the debris flow, including mobilization of landslide, in terms of the difference in topographical conditions, one for open slope and the other for channel. Moreover, both models are relatively simple and have great advantages, such as applicability to a wide area. As far as the reviewer knows, a similar approach has never been seen, so this is a highly novel research.

However, I consider that the paper is very long, does not focus on the novel aspects mentioned above, and the purpose, novelty, and important findings of the research are extremely difficult to recognize. I consider that the authors will make major revisions before accepting the paper for the journal.

Thank you for the helpful suggestions and comments. As originally written, the manuscript tells two stories in one. We use the common connection of zones of mobility to bring these two stories together. The result, as noted by reviewer 3, is that the manuscript is long, and the novelty of using different models for non-channelized versus channelized runout is not sufficiently highlighted.

We will revise the manuscript to shorten the main text and emphasize the novelty of the coupled model approach. We will achieve this goal by moving details related to the topographic analysis to a supplement and revising figure 5. In addition, we will refocus the discussion to highlight the benefits of the coupled model approach and mention of insightful aspects of the topographic analysis will place greater focus on the manuscript title of how topography controls mobility. Details of our revisions are provided in our response to comments 1, 2, and 3.

Here are detailed responses to comments (reviewer comments in grey and replies in black text):

1.  *Clarification of the significance of model coupling*

*The reviewer understands that the concept presented in Chapter 2 is the most important point of this study. However, it is difficult to say that the effectiveness of this concept has been sufficiently demonstrated and examined.*

*First, regarding the discussion section in Chapter 7, most of it (specifically Sections 7.2 and 7.3) is a discussion of the performance of individual models, which is not the main issue of this study. I*

*think that these sections may blur the focus and mask the significance of this study, so I propose deleting them. Anyway, please comment and consider the significance of the concept presented in Chapter 2 for disaster prediction, if the authors also consider the concept is the most important.*

*I consider that it is necessary to compare the results of the proposed model with those of the case where the H/L approach or debris flow growth approach is used alone without distinguishing between channel and open slope to be clear the role of model coupling. I also think that it is important to consider how the conditions for distinguishing between channel and open slope affect the results.*

*If the authors don't think other parts is more important, I hope the author clearly state it in Introduction.*

Thank you for recognizing the main highlight and pointing out the need for greater emphasis on the section 2 concepts of zones of mobility and the coupled model approach. Although our conclusion is focused on these highlights, other parts of the manuscript are long, and the discussion does not focus on the benefits of our modeling approach.

Section 7.2 was written to expand on the concepts presented in section 7.1 and discuss the influence of topography on the resultant calculation of mobility as measured by various metrics. We will combine the main point of section 7.2 with section 7.1 to allow more room for discussion of the main highlight. We will also revise and eliminate parts of section 7.3.

The results of a single versus coupled model approach are shown in earlier parts of the manuscript, but as noted, are not highlighted in the discussion. For example, figure 13 illustrates how the H/L approach is not adequate for estimation of inundation areas in the channel. Figures 16 and 17 show the contrasting results in highly dissected escarpments (fig. 16) versus open-slope topography (fig. 17), where no channels are present. The model results shown in these two areas highlight the benefits of the coupled model approach. We will add a new section to the discussion to bring the conversation back to these differences and provide greater emphasis on the benefits of two methods for delineation of runout zones (non-channelized and channelized). This addition to the discussion will emphasize the conceptual framework from section 2.1 and highlight the coupled model approach in the context of disaster prediction.

**２.The role of topographic analysis**

*The reviewer felt that the purpose/role of the topographic analysis in this study was not clear. The reviewer understood that the topographic analysis in this study is a preparatory step for creating a hazard(susceptibility) map, which is necessary to determine several empirical parameters of the proposed model. If this understanding is correct, I would like this point to be clarified. For example, the term "Topographic analysis" in Figure 5 (which I think is an important figure) could be called "Empirical parameters setting through topographic analysis" and the title of Section 4.2 could be "Topographic analysis for empirical parameters setting." I also think that the writing style and structure of Chapter 5 would be reconsider and rewritten to make it easier*

*to read and understand the role of the topographic analysis. I also think that the content of Section 7.1 needs to be reconsidered. In this case, Figure 5 would be revise easier to understand, for example, the linked-model part is set as the main (routine) in the figure, and empirical parameters setting through topographic analysis was added as a sub-part (routine).*

*On the other hand, if the topographic analysis itself was the purpose of this study, I think it is necessary to clarify in the introduction that there are issues that have not been clarified in previous studies of landslides and debris flow, and that what a kind of data is lacking. In fact, as the authors note in the Discussion section (e.g., 7.2), there is a great deal of prior research, and the reviewers could not consider that the results of the topographic analysis were significantly novel.*

We will provide more clarification on the purpose of the topographic analysis. This analysis was performed to obtain necessary parameters for the coupled model and justify the decision to apply the same parameters (growth zones, growth factors, maximum volumes, and H/L values) for all geologic terranes. Although the inventories were published, the extraction and analysis related to separate zones of mobility was novel. Likewise, the inventories were not specific to long runout debris flows, an important subset of the published datasets. Extraction of this subset (Maria's most mobile --MMM) was important to quantify constraining parameters in debris flow inundation zones. Statistics extracted from the runout zones of MMM (stream order, curvature, and $P_{src}$) are used in the modeling to define locations of potential debris flow growth. We acknowledge that the manuscript is long and given that the topographic analysis is not the primary highlight of the manuscript, we have moved details of this analysis to a supplement.

In section 2.2, we will add clarification that we further divided the landslide-affected area from the published inventories to distinguish channelized versus non-channelized runout. Major revisions to figure 5 (described below in comment #3) will place greater emphasis on the linked model.

By moving details of the topographic analysis to a supplement, the main manuscript will be shorter and more clearly highlight the coupled model approach. The revised discussion will briefly bring back the focus to insightful details revealed in the topographic analysis and then place greater focus on the coupled model.

*3. Clarifying overall model picture*

*It was very difficult to understand the overall structure of the proposed model. I think that Figure 5 fulfills that role, but it difficult to understand. I made a few comments about this figure in comment 2.*

*Also, I think it is easier to understand if the relationships among the three models are the same in Figures 2 and 5. Also, it would be easier to understand if it indicated where in the text the methods for each part of the figure are written.*

*Also, a list of the conditions and coefficients that need to be determined for the terrain classification and each model is helpful to understand overall structure of the proposed model,*

*Moreover. it would be easier for readers to understand if the determination method in this research was organized in the list. It would be very informative for future research if the authors could clarify which of the conditions that need to be determined have the greatest impact.*

Thank you for these comments. Reviewer 2 also indicates that figure 2 did not serve the intended purpose. Figure 2 was intended as a primer, showing a simplified version of the coupled model approach, without the details provided in figure 5. The three colored boxes in figure 2 and figure 5 are equivalent and highlight the main components of the coupled model. The use of slightly different wording in the two figures may have made this connection difficult to see. We will either eliminate or revise figure 2.

Figure 5 has been significantly revised and now includes the associated section numbers. The flowchart details describing the topographic analysis have been moved to a supplement. The revised figure 5 now shows the connection to the topographic analysis as a more simplified, minor component, providing a primary emphasis on the coupled model. The main components of the coupled model are now further highlighted with larger boxes and font size. Here is the revised figure 5 flowchart :

[Figure]

In addition, at the beginning of section 4, we will add a list of the necessary information for the coupled model approach.

*Minor comments*

*Fig. 8c   I couldn't distinguish between a plot of just the source and a plot of the entire area.*

We have revised the point and line symbols on this plot for additional clarity.

*L419 etc.  I found "area susceptible to shallow landslide" in several time in this section. However, in the table, the authors noted as "steep slope area". I hope the authors clarify it.*

On line 405, we state that these two areas are equivalent "…the percentage of study area susceptible to landslides, approximated as slopes > 30° ….". We then use both phrases interchangeably. For additional clarity, we will revise line 419 and all references after line 405 to use the term steep slopes.

*L429-432  It is hard to understand this part. Please add more explanation.*

We will revise and provide additional explanation for L429-432.

---

## Author Response (AR1)

**Authors reply and summary of revisions to 'Topographic controls on landslide mobility: Modeling hurricane-induced landslide runout and debris-flow inundation in Puerto Rico' based on reviewer comments**

by Dianne L. Brien, Mark E. Reid, Collin Cronkite-Ratcliff, and Jonathan P. Perkins

reviewer comments in grey and replies in black text.

**Response to Martin Mergili**

- The work flow of the study is, in principle, clearly presented and nicely illustrated through Fig. 5. There is just one point which is not clear to me: are the datasets used to develop the models and those used to evaluate their predictive success completely independent, or do they overlap? For H/L, it is written in L339 that, besides global datasets, also datasets from Hurricane Maria are used. Are those the same as used for evaluation (L373)? Are there similar issues for the debris flows? If yes, I suggest to address and justify such overlap in one or two sentences.

Thank you for the thoughtful comments and suggestions. Here is a reply:

**Input datasets for selection of model parameters**

For debris-flow inundation modeling, Maria's most mobile landslides (MMM) provided statistics to constrain the range of values for stream order, planform curvature, and $P_{src}$ (Table 7). Scenarios defining the location of debris flow growth were built using the extracted statistics in combination with stream slopes measured in the field. Debris-flow growth factors and maximum debris-flow volumes were not available for all MMM. The growth factors and volumes were obtained from Coe et al., 2021. The MMM dataset was used to test the predictive success of the debris flow inundation modeling.

For the landslide runout modeling, we explored a variety of published references with H/L values: 1) Hurricane Maria landslides (Bessette-Kirton, 2020), 2) worldwide datasets (Corominas, 1996), and 3) data from flume experiments (Iverson, 1997). Although one might expect that the ideal dataset is Hurricane Maria landslides, we found that median values presented in Bessette-Kirton, 2020 (27 to 34 degrees) would have resulted in an underestimation of runout and a gap between non-channelized runout zones and the channel. Maximum values from Hurricane Maria were from channelized flows – these values would be extremely low (14 degrees). It was apparent that H/L values extracted from regional datasets, such as the Hurricane Maria inventories, are influenced by the local slope and may not be a good metric for landslide mobility when used in this context. H/L is valuable for comparison of mobility when the local slope and hillslope lengths are equivalent. With much of the topography in Puerto Rico consisting of mountainous areas with greater than 40 degree slopes, landslides occurring on these steep slopes that do not make it to the bottom of the hill are common in the inventories and result in high values of α. The inventories included only a small number of landslides in non-channelized topography, hence we resorted to an approach that assesses how much the predicted area of runout is affected by changes in α. We ultimately conclude that 1) H/L values extracted from landslide inventories are strongly influenced by the local slope, and 2) for much of the terrane in Puerto Rico, there is very little difference in the area affected regardless of selected H/L value. Using the landslide source areas

published in the landslide inventories, figure 11 illustrates the minimal change in area affected (added yellow area) with a decrease of α from 25 to 20 degrees.

We attempt to explain the biases in H/L values extracted from regional landslide inventories from widespread landslide-inducing events in the results (6.1) and discussion (7.3.1). Ultimately, we did not quantitively assess predictive success given that in areas with predominantly steep slopes the statistics would have looked good, but this is really a bias in the distribution of topographic slopes rather than an indication of predictive success.

We have provided additional clarification in revisions to section 4.3.1 .

- The ROC analysis is, again, very well illustrated (Fig. 15) and explained. I am rather familiar with another type of ROC plot and was looking for values of the area under the curve (AUROC), which is very commonly used to evaluate the success of landslide simulations, but I found none. Would it be possible to use AUROC values, or is this just not useful for the type of ROC analysis performed here?

**ROC analysis**

AUROC values are typically used for comparison of different methods. In this case, one method is applied with different parameters rather than comparing different methods. The consideration of TPR allows direct comparison with the metric used for the source area modeling (Baum et al., 2024). For the intended purpose it is not as useful to extract AUROC values, since we need to identify a point in ROC space where TPR is less than 1.0 (TPR=1 is unobtainable without unrealistic over-prediction). With the goal of selecting parameters for two scenarios of regional susceptibility maps, PLR allows us to determine the balance between TPR and FPR for one method with different input parameters.

**Response to Anonymous Referee #2**

>>Thank you for the helpful comments and suggestions. We will especially work on revisions to figures. Here are replies to specific points.

Here are my specific points to consider:

- I understand that the potential landslide source areas are extracted from the literature. Still, I think it could be helpful to briefly give more details, perhaps in a paragraph, on how those potential source areas are calculated. This is to give the reader a bit more context without the need to go further to another publication.

We have added more details in section 3.1.4 to describe our collaborative work with Baum et al. (2024) using a slope stability analysis to identify potential landslide source areas. Here is the revised text:

*To estimate potential landslide source areas in our linked-model approach, we used areas identified by the combination of three USGS models: 1) REGOLITH (Baum et al., 2021), for soil-depth estimation, 2) TRIGRS (Baum et al., 2008), for pore-water pressures, and 3) Slabs3d (Baum et al., 2023), for quasi-three-dimensional (3D) slope-stability analysis. The areas susceptible to shallow landslides during prolonged, intense rainfall were defined by*

*factor of safety thresholds for high and very high susceptibility scenarios based on true positive rates compared to Hurricane Maria landslide inventories (Baum et al., 2024).*

Regarding the format, I believe there is a lot of potential to improve the illustrations. I understand it is a matter of taste, but with such quality of analysis, the figures do not do justice. Some of them have low resolution or are a bit convoluted. I would suggest using a similar illustration style (especially the maps). Specific comments are below:

- I do not see the purpose of Figure 2. It would be helpful if you could add more details or simply remove it and instead build the same connection you want to illustrate in Figure 5. That would also save a Figure since you already have quite a lot.

We removed figure 2.

- At the same time, the resolution of Figures 5, 6, and 7 urgently needs to be increased.

We will provide a version of these figures with increased resolution.

- In the legend of Fig 11, there is some underlined text that I suppose was not intended.

We removed the underlines from the legend text in Fig. 11.

- I would suggest removing the word 'perfect classification' from Figure 15.

We removed this text.

- There is a type in the first table. It should be Table 1 instead.

The table number has been corrected.

- Perhaps Fig 16 and 17 could be combined with a rectangle indicating the area being zoomed in.

Figures 16 and 17 are separate areas rather than a zoom in. The figure caption for figure 17 was revised for additional clarification regarding location.

**Response to Anonymous Referee #3**
Overall

This research is attempting to develop a model that combines the model for the landslide initiation and the sediment flow down as debris flow. In addition, two models are combined for the debris flow, including mobilization of landslide, in terms of the difference in topographical conditions, one for open slope and the other for channel. Moreover, both models are relatively simple and have great advantages, such as applicability to a wide area. As far as the reviewer knows, a similar approach has never been seen, so this is a highly novel research.

However, I consider that the paper is very long, does not focus on the novel aspects mentioned above, and the purpose, novelty, and important findings of the research are extremely difficult to recognize. I consider that the authors will make major revisions before accepting the paper for the journal.

Thank you for the helpful suggestions and comments. As originally written, the manuscript tells two stories in one. We use the common connection of zones of mobility to bring these two stories together. The result, as noted by reviewer 3, is that the manuscript is long, and the novelty of using different models for non-channelized versus channelized runout is not sufficiently highlighted.

We have revised the manuscript to shorten the main text and emphasize the novelty of the linked model approach. We have moved details related to the topographic analysis to a supplement and revised figure 4 (formerly figure 5). In addition, we added a section to the discussion to explain the benefits of the linked model approach and more clearly highlight aspects of the topographic analysis with greater focus on the manuscript title of how topography controls mobility. Details of our revisions are provided in our response to comments 1, 2, and 3.

1. Clarification of the significance of model coupling

The reviewer understands that the concept presented in Chapter 2 is the most important point of this study. However, it is difficult to say that the effectiveness of this concept has been sufficiently demonstrated and examined.

First, regarding the discussion section in Chapter 7, most of it (specifically Sections 7.2 and 7.3) is a discussion of the performance of individual models, which is not the main issue of this study. I think that these sections may blur the focus and mask the significance of this study, so I propose deleting them. Anyway, please comment and consider the significance of the concept presented in Chapter 2 for disaster prediction, if the authors also consider the concept is the most important.

I consider that it is necessary to compare the results of the proposed model with those of the case where the H/L approach or debris flow growth approach is used alone without distinguishing between channel and open slope to be clear the role of model coupling. I also think that it is important to consider how the conditions for distinguishing between channel and open slope affect the results.

If the authors don't think other parts is more important, I hope the author clearly state it in Introduction.

Thank you for recognizing the main highlight and pointing out the need for greater emphasis on the section 2 concepts of zones of mobility and the linked model approach. Although our conclusion is focused on these highlights, other parts of the manuscript are long, and the discussion does not focus on the benefits of our modeling approach.

In the discussion, we have combined the main point of section 7.2 with a revised section 7.1.

The results of a single versus linked model approach are shown in earlier parts of the manuscript, but as noted, are not highlighted in the discussion. For example, figure 11 illustrates how the H/L approach is not adequate for estimation of inundation areas in the channel. Figures 14 and 15 show the contrasting

results in highly dissected escarpments (fig. 14) versus open-slope topography (fig. 15), where no channels are present. The model results shown in these two areas highlight the benefits of the linked model approach. We have added a new section to the discussion (7.2) to bring the conversation back to these differences and provide greater emphasis on the benefits of two methods for delineation of runout zones (non-channelized and channelized). This addition to the discussion emphasizes the conceptual framework from section 2.1 and highlights the linked model approach in the context of disaster prediction.

*2. The role of topographic analysis*

*The reviewer felt that the purpose/role of the topographic analysis in this study was not clear. The reviewer understood that the topographic analysis in this study is a preparatory step for creating a hazard(susceptibility) map, which is necessary to determine several empirical parameters of the proposed model. If this understanding is correct, I would like this point to be clarified. For example, the term "Topographic analysis" in Figure 5 (which I think is an important figure) could be called "Empirical parameters setting through topographic analysis" and the title of Section 4.2 could be "Topographic analysis for empirical parameters setting." I also think that the writing style and structure of Chapter 5 would be reconsider and rewritten to make it easier to read and understand the role of the topographic analysis. I also think that the content of Section 7.1 needs to be reconsidered. In this case, Figure 5 would be revise easier to understand, for example, the linked-model part is set as the main (routine) in the figure, and empirical parameters setting through topographic analysis was added as a sub-part (routine).*

*On the other hand, if the topographic analysis itself was the purpose of this study, I think it is necessary to clarify in the introduction that there are issues that have not been clarified in previous studies of landslides and debris flow, and that what a kind of data is lacking. In fact, as the authors note in the Discussion section (e.g., 7.2), there is a great deal of prior research, and the reviewers could not consider that the results of the topographic analysis were significantly novel.*

We have provided additional clarification on the purpose of the topographic analysis. This analysis was performed to obtain necessary parameters for the linked model and justify the decision to apply the same parameters (growth zones, growth factors, maximum volumes, and H/L values) for all geologic terranes. Although the inventories were published, the extraction and analysis related to separate zones of mobility was novel. Likewise, the inventories were not specific to long runout debris flows, an important subset of the published datasets. Extraction of this subset (Maria's most mobile --MMM) was important to quantify constraining parameters in debris flow inundation zones. Statistics extracted from the runout zones of MMM (stream order, curvature, and $P_{src}$) are used in the modeling to define locations of potential debris flow growth. We acknowledge that the manuscript is long and given that the topographic analysis is not the primary highlight of the manuscript, we have moved details of this analysis to a supplement.

In section 2.2, we added clarification that we further divided the landslide-affected area from the published inventories to distinguish channelized versus non-channelized runout. Major revisions to figure 5 (figure 4 in revised version) place greater emphasis on the linked model.

By moving details of the topographic analysis to a supplement, the main manuscript is shorter and more clearly highlights the linked model approach. The revised discussion briefly brings the focus back to

insightful details revealed in the topographic analysis and the title of the manuscript, and then places greater focus on the linked model.

*3. Clarifying overall model picture*

*It was very difficult to understand the overall structure of the proposed model. I think that Figure 5 fulfills that role, but it difficult to understand. I made a few comments about this figure in comment 2.*

*Also, I think it is easier to understand if the relationships among the three models are the same in Figures 2 and 5. Also, it would be easier to understand if it indicated where in the text the methods for each part of the figure are written.*

*Also, a list of the conditions and coefficients that need to be determined for the terrain classification and each model is helpful to understand overall structure of the proposed model, Moreover. it would be easier for readers to understand if the determination method in this research was organized in the list. It would be very informative for future research if the authors could clarify which of the conditions that need to be determined have the greatest impact.*

Thank you for these comments. Reviewer 2 also indicates that figure 2 did not serve the intended purpose. Figure 2 was intended as a primer, showing a simplified version of the linked model approach, without the details provided in figure 5. The three colored boxes in figure 2 and figure 5 (figure 4 in revised version) are equivalent and highlight the main components of the linked model. The use of slightly different wording in the two figures may have made this connection difficult to see. We have eliminated figure 2.

Figure 5 has been significantly revised and now includes the associated section numbers. The flowchart details describing the topographic analysis have been moved to a supplement. The revised figure 5 now shows the connection to the topographic analysis as a more simplified, minor component, providing a primary emphasis on the linked model. The main components of the linked model are now further highlighted with larger boxes and font size. Here is the revised figure 5 flowchart :

[Figure]

In addition, at the beginning of section 4.3, we added a new table, listing the necessary information for the linked model approach.

*Minor comments*

*Fig. 8c  I couldn't distinguish between a plot of just the source and a plot of the entire area.*

We have revised the point and line symbols on Fig 8c (figure 7c in revised version) for additional clarity.

*L419 etc. I found "area susceptible to shallow landslide" in several time in this section. However, in the table, the authors noted as "steep slope area". I hope the authors clarify it.*

On line 405, we state that these two areas are equivalent "…the percentage of study area susceptible to landslides, approximated as slopes > 30° ….". We have revised the text to always use the same terminology thereafter.

*L429-432 It is hard to understand this part. Please add more explanation.*

We have revised the text in L429-432.